# The NuRD component CHD3 promotes BMP signalling during cranial neural crest cell specification

Zoe H Mitchell [ID][1], Joery den Hoed [ID][2], Willemijn Claassen [ID][2], Martina Demurtas[1], Laura Deelen[1], Philippe M Campeau [ID][3,4], Karen Liu [ID][5], Simon E Fisher [ID][2,6] & Marco Trizzino [ID][1✉]

## Abstract

**Pathogenic genetic variants in the NuRD component *CHD3* cause Snijders Blok–Campeau Syndrome, a neurodevelopmental disorder manifesting with intellectual disability and craniofacial anomalies. To investigate the role of CHD3 in craniofacial development, we differentiated control and *CHD3*-depleted human-induced pluripotent stem cells into cranial neural crest cells (CNCCs). In control lines, CHD3 is upregulated in early stages of CNCC specification, where it enhances the BMP signalling response by opening chromatin at BMP-responsive *cis*-regulatory elements and by increasing expression of BMP-responsive transcription factors, including DLX paralogs. CHD3 loss leads to repression of BMP target genes and loss of chromatin accessibility at *cis*-regulatory elements usually bound by BMP-responsive factors, causing an imbalance between BMP and Wnt signalling. Consequently, the CNCC specification fails, replaced by aberrant early-mesoderm identity, which can be partially rescued by titrating Wnt levels. Our findings highlight a novel role for CHD3 as a pivotal regulator of BMP signalling, essential for proper neural crest specification and craniofacial development. Moreover, these results suggest a molecular mechanism for the craniofacial anomalies of Snijders Blok–Campeau Syndrome.**

**Keywords** NuRD; CHD3; Cranial Neural Crest; Snijders Blok–Campeau Syndrome; BMP
**Subject Categories** Chromatin, Transcription & Genomics; Development; Signal Transduction

## Introduction

Human development is a highly complex process which depends upon the precise regulation of gene expression to determine cell fates. This regulation relies on a series of epigenetic mechanisms, including histone modifications, DNA methylation, and chromatin remodelling (Jaenisch and Bird 2003; Yadav et al, 2018).

One key chromatin regulator is the nucleosome remodelling and deacetylase (NuRD) complex. The NuRD complex is an ATP-dependent complex that possesses both histone deacetylation and nucleosome remodelling activity (Zhang et al, 1998; Xue et al, 1998a; Tong et al, 1998; Low et al, 2016; Zhang et al, 2016). While NuRD was initially thought to act as a repressor of gene expression (Zhang et al, 1998; Xue et al, 1998a; Tong et al, 1998), recent evidence has demonstrated that this complex is able to both repress and activate transcription of target genes through the reorganisation of nucleosome structure (Zhang et al, 2012; Miccio et al, 2010; Bornelöv et al, 2018). NuRD mediated regulation of gene expression is thought to play a key role during development (Kaji et al, 2006; Hoffmann and Spengler, 2019) and is driven by the nucleosome remodelling activity of the complex, which is provided by one of three mutually exclusive CHD subunits: CHD3, CHD4 or CHD5 (Bornelöv et al, 2018).

The CHD proteins contain an ATPase/helicase domain and a chromodomain motif, which enable the alteration of chromatin structure (Woodage et al, 1997). Each NuRD complex only harbours a single CHD protein, with different CHD-NuRD complexes displaying distinct functions and targeting distinct sets of genes (Hoffmeister et al, 2017). It has further been suggested that different NuRD configurations may provide time-, tissue- and context-dependent function (Basta and Rauchman, 2015; Thompson et al, 2003; Zhuang et al, 2014). For example, a study on mouse cortical development showed that different CHDs were incorporated in the NuRD complex at different developmental stages, with each stage-specific NuRD complex displaying distinct functions (Nitarska et al, 2016). Given their unique roles, it is perhaps unsurprising that loss or mutation of each one of the CHD proteins results in specific neurodevelopmental disorders.

One example is Snijders Blok–Campeau syndrome, a rare, autosomal dominant, neurodevelopmental disorder resulting exclusively from pathogenic variants within *CHD3* (Snijders Blok et al, 2018), with affected individuals presenting with a variety of variants, including heterozygous missense variants within the ATPase/helicase domain, and, less frequently, heterozygous loss-of-function variants (Snijders Blok et al, 2018; Drivas et al, 2020; van der Spek et al, 2022). It has been hypothesised that the missense variants may alter the chromatin remodelling ability of CHD3,

[1]Department of Life Sciences, Imperial College London, London, UK. [2]Language and Genetics Department, Max Planck Institute for Psycholinguistic, Nijmegen, The Netherlands. [3]CHU Sainte-Justine Research Center, Montreal, QC, Canada. [4]Department of Pediatrics, University of Montreal, Montreal, QC, Canada. [5]Centre for Craniofacial and Regenerative Biology, King's College London, London, UK. [6]Donders Institute for Brain, Cognition, and Behaviour, Raboud University, Nijmegen, The Netherlands.
✉E-mail: m.trizzino@imperial.ac.uk

which could represent a potential pathogenic mechanism behind Snijders Blok–Campeau syndrome (Snijders Blok et al, 2018).

Affected individuals present with a broad and variable phenotype including different degrees of intellectual disability, impaired speech and language, and macrocephaly (Snijders Blok et al, 2018; Drivas et al, 2020; van der Spek et al, 2022), along with distinct facial anomalies including a broad, bossed forehead, widely spaced and deep-set eyes, narrow palpebral fissures, midface hypoplasia and low-set ears (Snijders Blok et al, 2018; Drivas et al, 2020; van der Spek et al, 2022). So far, only two individuals have been identified with a potential pathogenic *CHD3* variant in both copies of the gene, and specifically a homozygous in-frame insertion (c.5384_5389dup; p.Arg1796_Phe1797insTrpArg) (Goldfarb Yaacobi and Sukenik Halevy 2024). These individuals were reported to display a more severe phenotype than cases carrying heterozygous variants, including more distinct facial dysmorphism and severe intellectual disability (Goldfarb Yaacobi and Sukenik Halevy, 2024). The distinct facial phenotype observed in individuals with pathogenic *CHD3* variants suggests that this NuRD subunit may play an important role in craniofacial development, but this has not been investigated so far.

Craniofacial development is underpinned by the cranial neural crest cells (CNCCs), which constitute an embryonic multipotent cell type from which the bones, cartilage and connective tissues of the face are formed (Vega-Lopez et al, 2018). CNCCs are generated in the dorsal portion of the neural tube, at the border between the neural plate and the surface ectoderm (Ruffins and Bronner-Fraser, 2000). Following neural crest induction, these cells undergo epithelial-to-mesenchymal transition (EMT) and subsequently migrate and populate the relevant regions of the developing embryo, where they differentiate into different derivatives, including the craniofacial bones and cartilage (Cordero et al, 2011).

The process of CNCC specification and formation is complex, and requires the coordinated activity of multiple signalling pathways, key among which are the BMP, and Wnt pathways (Stuhlmiller and García-Castro, 2012).

The BMP proteins bind BMP receptors to activate SMAD proteins, which enter the nucleus to trigger specific gene expression programmes (Kishigami and Mishina, 2005). This is mediated by specific BMP-responsive transcription factors, including paralogs of the DLX and MSX families during patterning of the facial mesenchyme (Nishimura et al, 2012; Rahman et al, 2015; Mishina and Snider, 2014). On the other hand, Wnt signalling is required at multiple stages, with roles in neural crest induction, specification, and subsequent migration and differentiation. Wnt ligands bind to Frizzled receptors, allowing β-catenin to enter the nucleus and activate transcription (Logan and Nusse, 2004). A finely tuned balance of Wnt, BMP and FGF signalling is required throughout craniofacial development to enable neural crest induction, specification, migration and subsequent cell fate determination (García-Castro et al, 2002; Maj et al, 2016; Lee et al, 2004; Roth et al, 2021; Liao et al, 2022; Correia et al, 2007; Kanzler et al, 2000; Bonilla-Claudio et al, 2012; Stuhlmiller and García-Castro, 2012). We hypothesise that factors such as CHD3 establish appropriate CNCC-chromatin state allowing synchronisation of signalling pathways during lineage specification.

The craniofacial anomalies seen in individuals with Snijders Blok–Campeau syndrome suggest that CHD3 is essential for the specification and/or differentiation of CNCCs. In this study, we therefore sought to establish the role of CHD3 in craniofacial development using human iPSC models with either heterozygous or homozygous frameshift variants that result in loss of expression of the allele/gene. Importantly, established protocols are available to differentiate iPSCs into migratory CNCCs (Prescott et al, 2015; Pagliaroli et al, 2021; Barnada et al, 2024b). With this approach, we found that CHD3 is required to allow response to BMP during the specification of the CNCCs. Namely, CHD3 regulates accessibility at enhancers bound by BMP-responsive transcription factors, including DLX and MSX families, as well as the expression of these factors. In the absence of CHD3, BMP response is not effective, and this leads to a Wnt/BMP imbalance. Thus, CNCC specification fails, replaced by mesodermal identity, which can be partially rescued by titrating Wnt levels.

# Results

## CHD3 is not required for the pluripotent identity of the iPSCs

To investigate the role of CHD3 in CNCC specification, we used heterozygous and homozygous *CHD3* knockout iPSC lines (and isogenic controls) generated in a companion study (preprint: den Hoed et al, 2024) by means of CRISPR/Cas9 gene-editing in the BIONi010-A iPSC line (Fig. 1A). Specifically, for this study, we used two different homozygous clones (hereafter *CHD3*-KO clones 1 and 2) in which Cas9 independently targeted the third exon of the *CHD3* gene, producing a 1-base deletion (c.298delG), which generated a premature stop codon downstream (Fig. 1B) (preprint: den Hoed et al, 2024). Moreover, we used two heterozygous clones (hereafter *CHD3*-HET-KO clones 1 and 2) that were generated using the same targeting strategy as the homozygous KO, that carried c.298insA and c.298insT variants respectively (preprint: den Hoed et al, 2024).

In the original paper in which these CRISPR lines were generated (preprint: den Hoed et al, 2024), it was already established that neither the *CHD3*-KO nor the *CHD3*-HET-KO affects the pluripotency of the iPSCs. We further corroborated this finding in the present study with additional experiments. Specifically, RT-qPCR revealed no significant difference in the expression of the main pluripotency markers (*NANOG, OCT4* and *SOX2*) when comparing the *CHD3*-KO and *CHD3*-WT iPSCs (Appendix Fig. S1A). As expected, expression of *CHD3* was significantly lower in *CHD3*-KO iPSCs compared to *CHD3*-WT (Appendix Fig. S1B). Furthermore, a significant reduction of CHD3 at the protein level in *CHD3*-HET-KO iPSCs and a complete loss of CHD3 at the protein level in the *CHD3*-KO iPSCs was also observed, confirming the success of the CRISPR knockout (Appendix Fig. S1C) (preprint: den Hoed et al, 2024). RNA-seq performed at the iPSC stage found only 25 genes to be differentially expressed between *CHD3*-WT and *CHD3*-HET-KO iPSCs, and 62 genes to be differentially expressed between *CHD3*-WT and *CHD3*-KO iPSCs (FDR < 5%; logFC > 1.5 or < −1.5). Of these differentially expressed genes, 17 genes were unique between *CHD3*-WT and *CHD3*-HET-KO iPSCs and 54 genes were unique between *CHD3*-WT and *CHD3*-KO iPSCs, with 8 genes appearing in both analyses. None of the known pluripotency factors were differentially expressed (Dataset EV1). Immunofluorescence and flow cytometry

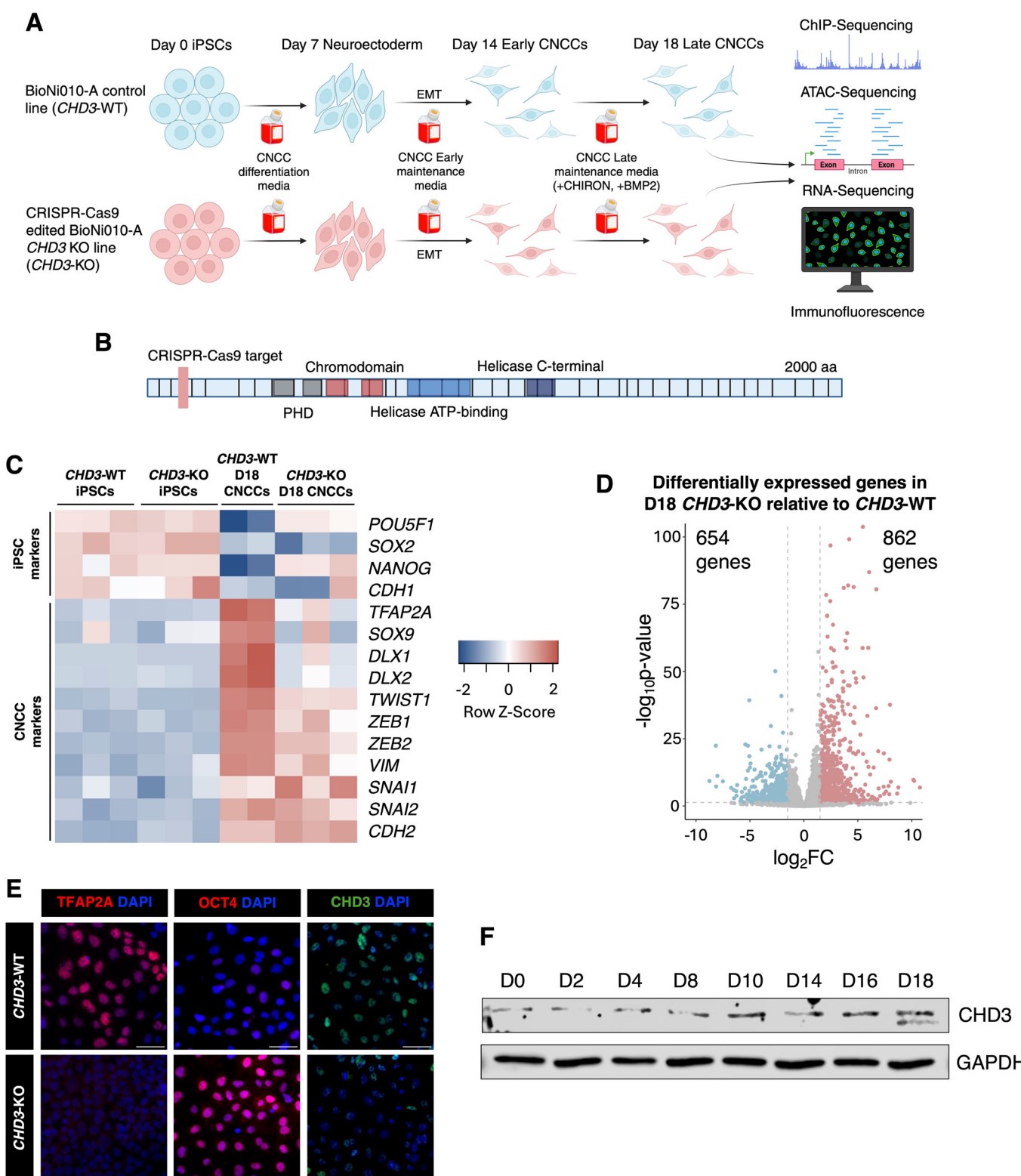

analysis of several key pluripotency markers confirmed that protein levels of pluripotency factors are not affected by CHD3 loss (Appendix Fig. S1D,E). Overall, these data suggest that CHD3 loss has a modest impact on the iPSC transcriptome, and it does not

affect the pluripotent gene and protein network. Consistent with this, *CHD3*-KO iPSCs exhibited regular morphology, forming tightly packed colonies with well-defined edges (Appendix Fig. S1F). To further corroborate the pluripotent state of the *CHD3*-KO

**Figure 1. Loss of CHD3 impairs CNCC specification.**

(A) A graphical illustration of the experimental pipeline. Made with biorender.com. (B) A schematic of the CHD3 gene indicating the site targeted by CRISPR-Cas9. Human CHD3 isoform 1, NM_001005273.2, 2000 aa, 40 exons (canonical). Made with biorender.com. (C) Heatmap displaying expression of key pluripotency markers and CNCC markers in CHD3-WT iPSCs, CHD3-KO iPSCs, CHD3-WT day-18 CNCCs and CHD3-KO day-18 CNCCs. (n = 3 biological replicates for each condition except CHD3-WT day-18 CNCCs where n = 2 biological replicates). (D) Volcano plot of differentially expressed genes in CHD3-KO relative to CHD3-WT in day-18 CNCCs (n = 3 biological replicates). Blue dots represent downregulated genes with P-adj < 0.05 and $\log_2$FoldChange < −1.5. Red dots represent upregulated genes with P-adj < 0.05 and $\log_2$FoldChange > 1.5. Adjusted P values were calculated by DESeq2 (Love et al, 2014) using the Wald test. (E) Immunofluorescence for CNCC marker TFAP2A, pluripotency marker OCT4, and CHD3 in CHD3-WT and CHD3-KO day-18 CNCCs. Scale bar: 50 μm. (F) Time-course western blot for CHD3 in CHD3-WT during the course of differentiation from iPSCs to CNCCs. GAPDH was used as a loading control. Source data are available online for this figure.

iPSCs, we performed trilineage differentiation and found that CHD3-KO iPSCs were able to successfully differentiate into all three germ layers (Appendix Fig. S1G).

In summary, these data confirm that the CRISPR knockout of CHD3 was successful and that loss of CHD3 has no impact on iPSC pluripotent identity.

## Loss of CHD3 impairs CNCC specification

Next, we investigated whether CHD3-KO affects CNCC specification. To achieve this, we leveraged an established protocol (Prescott et al, 2015; Bajpai et al, 2010), which has been previously adapted by our lab (Pagliaroli et al, 2021; Barnada et al, 2024b). With this protocol, fully specified, migratory CNCCs are generated in 18 days.

We first tested the CNCC specification protocol on the CHD3-WT clones. At the endpoint of the differentiation (day 18), the cells expressed genes typical of CNCC identity (e.g., SOX9, TFAP2A, TWIST1, NR2F1, SNAI1/2; Fig. 1C), along with markers of EMT and mesenchymal state (e.g., VIM, ZEB2, SNAI1/2, CDH2; Fig. 1C). Conversely, epithelial and pluripotency markers were downregulated (e.g., CDH1, POU5F1, NANOG, SOX2; Fig. 1C). Overall, these data confirmed that both clones of CHD3-WT iPSCs were able to successfully differentiate into migratory CNCCs.

We went on to investigate whether CHD3 loss had an impact on CNCC specification. To this end, we differentiated the CHD3-WT, CHD3-HET-KO, and CHD3-KO clones into CNCCs and conducted RNA-seq, paired with immunofluorescence for CNCC and iPSC markers. The cells were collected at the endpoint of the protocol (day 18). Comparing the two CHD3-WT clones with the CHD3-HET-KO counterparts, we found that heterozygous loss of CHD3 did not have a major impact on CNCC specification. In fact, only 36 genes were differentially expressed in CHD3-HET-KO CNCCs relative to the CHD3-WT lines (FDR < 5%; logFC > 1.5 or < −1.5; Appendix Fig. S2; Dataset EV2). This could potentially be due to compensation from the wild-type allele.

In stark contrast to the subtle effects of a heterozygous loss of CHD3, 1516 genes were differentially expressed when comparing CHD3-KO and CHD3-WT CNCCs (FDR < 5%; logFC > 1.5 or < −1.5; Fig. 1D; Appendix Fig. S3). Of these, 862 genes were upregulated in the CHD3-KO CNCCs, while 654 were downregulated. Many CNCC markers were downregulated in CHD3-KO, including TFAP2A, TWIST1, SOX9 and NR2F1 (Fig. 1C). On the other hand, pluripotency genes such as POU5F1 (OCT4) and NANOG were upregulated (Fig. 1C; Dataset EV3). TFAP2A downregulation and OCT4 upregulation in CHD3-KO CNCCs were also detected at the protein level (Fig. 1E). In addition, we observed a substantial reduction in CHD3 protein level in the CHD3-KO CNCCs compared to the CHD3-WT CNCCs (Fig. 1E).

Finally, we noted that the two other NuRD paralogs CHD4 and CHD5 were not upregulated in CHD3-KO CNCCs, potentially excluding compensatory mechanisms between NuRD subunits.

Together, these data suggest that CHD3 has an important role in CNCC specification. This role is reflected by the gradual upregulation of this protein, from relatively low levels in iPSCs and during neuroectoderm formation (~days 1–8; Fig. 1F), to higher levels in pre-migratory and migratory CNCCs (days 10–18; Fig. 1F). The CHD3 protein upregulation reflects a progressive upregulation in CHD3 gene expression as suggested by RNA-seq performed in iPSCs (CHD3 median TPM = 3.4), early CNCCs (day 14, median TPM = 11.5) and late, fully specified CNCCs (median TPM = 18.8). Interestingly, a slightly shorter isoform of CHD3 (ENST00000358181.8), lacking the original exons 1 (replaced with an alternative exon 1) and 33 is expressed at relatively low levels in fully specified CNCCs (day-18 median TPM = 3.3), while it is not expressed at the previous time points. The expression of the shorter isoform at day 18 is also observable at the protein level (Fig. 1F). This upregulation of CHD3 during the later stages of CNCC specification may also explain the trace levels of CHD3 protein observed in the CHD3-KO CNCCs. Consistent with the upregulation of CHD3 during the latter stages of CNCC specification, time-course RT-qPCR experiments revealed that the decreased expression of CNCC markers in CHD3-KO relative to CHD3-WT was observed from ~day 10 onwards (Appendix Fig. S4), which mirrors CHD3 upregulation at the same timepoint. Similarly, reduced expression of the mesenchymal marker VIM and increased expression of the epithelial marker EPCAM in CHD3-KO were also observed at the same stage (Appendix Fig. S4). This suggests that the observed defects in CNCC specification mirror the timing of CHD3 upregulation during this developmental process, further supporting a causal relationship.

## CHD3-KO cells undergo mesodermal fate

To shed light on the function of CHD3 in CNCC specification, we performed Gene Ontology (GO) analysis on the 1516 genes differentially expressed in CHD3-KO CNCCs on day 18.

GO terms downregulated in the CHD3-KO CNCCs were mainly associated with development, morphogenesis, patterning, and cell motility (Fig. 2A). Conversely, the upregulated genes were enriched for cell-cell junction and ion channel terms, potentially explaining the impaired EMT process and the persistent epithelial state of the CHD3-KO cells (Fig. 2B). However, scratch assay indicated that the CHD3-KO CNCCs exhibited only a mild impairment in migration with a slight decrease in migration over 6 h compared to the CHD3-WT CNCCs and no significant difference over 24 h (Appendix Fig. S5).

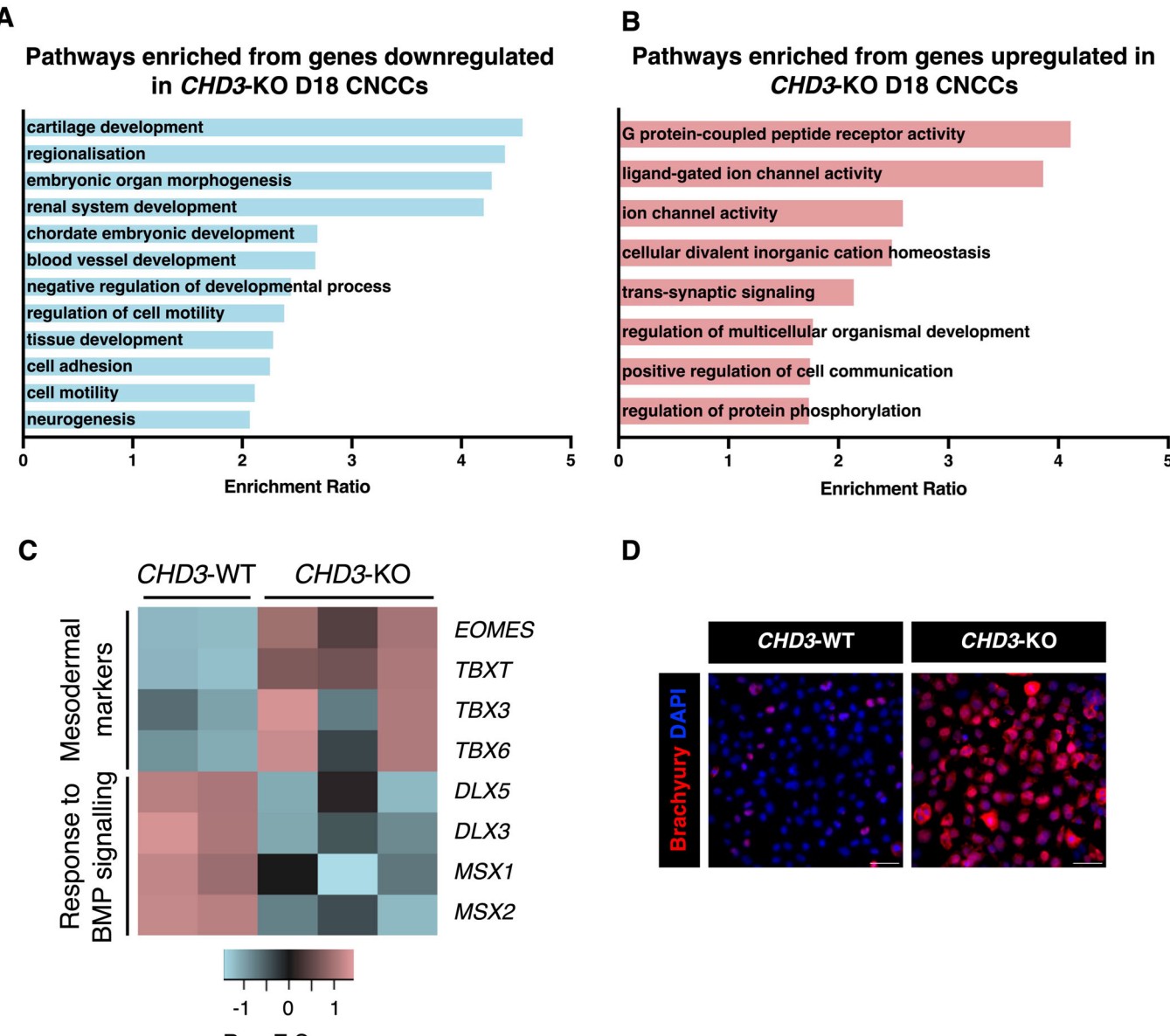

**Figure 2. CHD3-KO CNCCs display aberrant expression of early mesodermal markers.**

(A, B) GO term analysis of (A) significantly downregulated and (B) significantly upregulated genes in day-18 CHD3-KO CNCCs as determined using WebGestalt pathway analysis. (C) Heatmap displaying expression of mesodermal markers and genes involved in BMP signalling in CHD3-WT and CHD3-KO day-18 CNCCs. (D) Immunofluorescence for the mesodermal marker brachyury in CHD3-WT and CHD3-KO D18 CNCCs. Scale bar: 50 µm. Source data are available online for this figure.

Importantly, the CHD3-KO cells also exhibited upregulation of genes typically expressed in the primitive streak and in the early pre-migratory mesoderm, such as EOMES, TBXT, TBX3, TBX6, and MIXL1, paired with downregulation of BMP-responsive transcription factors, including several paralogs of the DLX and MSX families (Fig. 2C,D) (Luo et al, 2001; Zhu et al, 2023; Mishina and Snider, 2014).

## Chromatin accessibility is impaired at BMP-responsive cis-regulatory elements

Our results so far indicate that the CHD3-KO cells display an aberrant early-mesoderm signature, while failing to upregulate

genes associated with BMP signalling, developmental patterning and CNCC identity. Next, we sought to elucidate the underlying molecular mechanism.

CHD3 is part of the NuRD chromatin remodelling complex, therefore we reasoned that the observed phenotypes could be caused by dysregulation of chromatin accessibility. To test this, we performed ATAC-seq on CHD3-WT and CHD3-KO clones at the endpoint of the CNCC specification protocol (day 18). We identified 23,121 ATAC-seq peaks which were conserved between all CHD3-WT and CHD3-KO clones (FDR < 5%; Fig. 3A–C; Appendix Fig. S6). On the other hand, 17,543 peaks were found exclusively in the CHD3-WT CNCCs but not in CHD3-KO

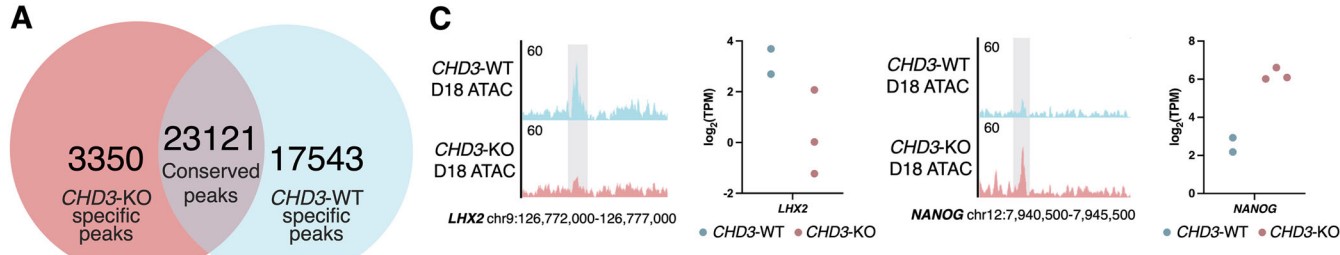

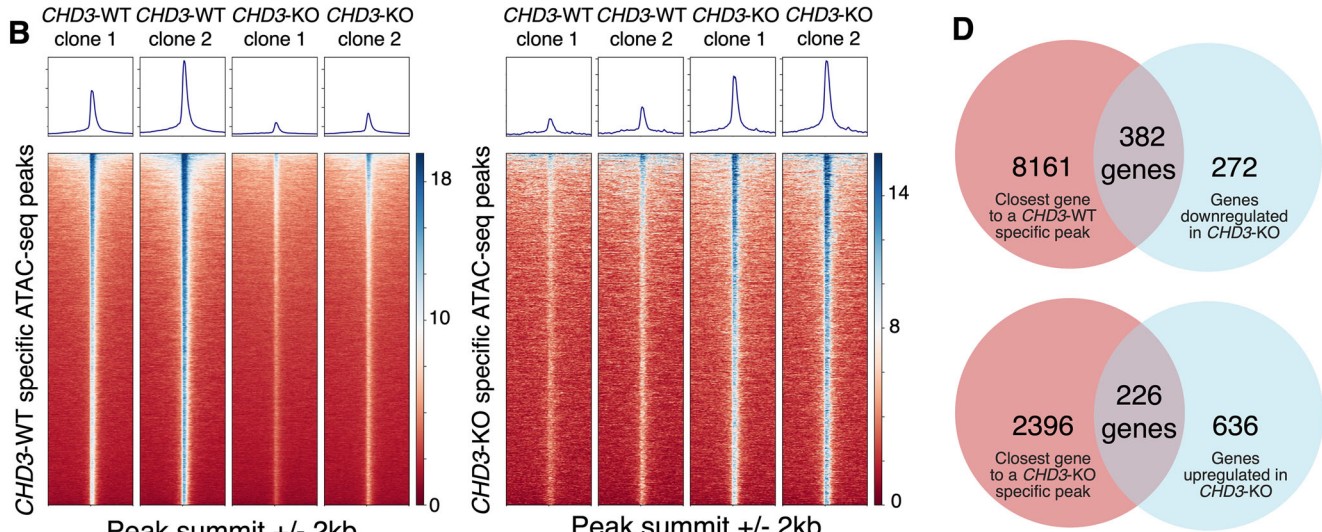

## E
### CHD3-KO specific D18 ATAC-seq peaks

| Transcription Factor | Motif | P-value |
|---|---|---|
| EOMES | | 1e-53 |
| TBXT | | 1e-53 |
| MAFF | | 1e-43 |
| GATA1 | | 1e-38 |
| GATA2 | | 1e-37 |
| TBX6 | | 1e-36 |
| OCT4 | | 1e-34 |
| GATA6 | | 1e-33 |
| GATA3 | | 1e-31 |

## F
### CHD3-WT specific D18 ATAC-seq peaks

| Transcription Factor | Motif | P-value |
|---|---|---|
| TFAP2A | | 1e-87 |
| TFAP2C | | 1e-73 |
| OTX2 | | 1e-43 |
| GSC | | 1e-41 |
| GRHL2 | | 1e-39 |
| NR2F6 | | 1e-30 |
| MSX/DLX5 | | 1e-29 |
| NR2F2 | | 1e-29 |
| MSX/DLX3 | | 1e-29 |

**Figure 3. CHD3 loss alters chromatin accessibility in D18 CNCCs.**

(A) Venn diagram showing the number of CHD3-KO specific ATAC-seq peaks, CHD3-WT specific ATAC-seq peaks and conserved (peaks present in both CHD3-WT and CHD3-KO) ATAC-seq peaks in day-18 CNCCs. (B) Heatmaps showing ATAC-seq peaks present in individual CHD3-WT and CHD3-KO biological replicates which are either CHD3-WT or CHD3-KO specific in day-18 CNCCs. (C) Example of CHD3-WT-specific and CHD3-KO-specific ATAC-seq peaks, present at the LHX2 locus and NANOG locus, respectively, visualised in UCSC genome browser and corresponding dot plots produced using RNA-seq data from day-18 CHD3-WT (n = 2 biological replicates) and CHD3-KO (n = 3 biological replicates) CNCCs. Log$_2$(TPM) for LHX2 and NANOG are displayed. (D) Venn diagrams showing the number of genes which are both closest to a CHD3-WT-specific ATAC-seq peak at day 18 and downregulated in CHD3-KO at day 18 or are closest to a CHD3-KO-specific ATAC-seq peak at day 18 and upregulated in CHD3-KO at day 18. (E, F) Tables of motifs enriched in either (E) CHD3-KO-specific day-18 ATAC-seq peaks or (F) CHD3-WT-specific D18 ATAC-seq peaks identified using HOMER. Source data are available online for this figure.

counterparts (hereafter, *CHD3*-WT-specific peaks), while 3350 peaks were found to be exclusive to the *CHD3*-KO CNCCs (hereafter *CHD3*-KO specific peaks; Fig. 3A–C; Appendix Fig. S6).

Overall, 90% of the *CHD3*-WT-specific and 97% of the *CHD3*-KO-specific ATAC-seq peaks were distal from the nearest Transcription Start Site (TSS; i.e., distance >1 Kb), suggesting that most of these regions are putative enhancers. Hence, these results indicate that loss of CHD3 had a significant impact on chromatin accessibility at distal *cis*-regulatory elements, both due to an overall reduction in chromatin accessibility in the *CHD3*-KO along with aberrant changes to the accessible sites.

Notably, 382 of the 654 genes downregulated in *CHD3*-KO cells (58%) were also the nearest gene to an enhancer that lost accessibility in these cells (i.e., *CHD3*-WT-specific ATAC-seq peak), while 226 of the 862 upregulated genes (26%) were also the closest gene to an enhancer that gained accessibility (Fig. 3D). This pattern suggests that CHD3 regulates a significant number of genes through direct regulation of chromatin accessibility at their *cis*-regulatory elements, acting both as activator and repressor, consistent with recent studies that suggested that the NuRD complex can both activate and repress gene expression in a context specific manner (Zhang et al, 2012; Miccio et al, 2010; Bornelöv et al, 2018).

Motif analysis of the *cis*-regulatory elements aberrantly active in *CHD3*-KO cells showed enrichment for binding motifs of several primitive streak and early-mesoderm transcription factors, including EOMES, TBXT (BRACHYURY), TBX3/6, and GATA (Fig. 3E) (Sadahiro et al, 2018; Faial et al, 2015; Costello et al, 2011; Lentjes et al, 2016). On the other hand, the *cis*-regulatory elements aberrantly repressed in *CHD3*-KO cells were enriched for motifs of transcription factors implicated in CNCC specification (TFAP2A/C, NR2F2/6), BMP response (DLX/MSX families and OTX2), and patterning (Fig. 3F) (Nishimura et al, 2012; Rahman et al, 2015; Mishina and Snider, 2014; Gammill and Sive, 2000; Luo et al, 2001).

In summary, both the RNA-seq and the ATAC-seq consistently demonstrated that the CHD3-deficient cells induced an aberrant mesodermal programme, while failing to activate CNCC specification possibly due to impaired response to BMP signalling.

## CHD3 primes BMP response in the developing CNCCs

Our experiments conducted in terminally specified CNCCs (day 18) indicated that BMP signalling response might be dysregulated in *CHD3*-KO conditions and that the cells acquire an unexpected mesodermal signature. It has previously been established that a combination of Wnt and FGF promotes the differentiation of iPSCs into mesodermal lineages (Turner et al, 2014; Amel et al, 2023; Sudheer et al, 2016; Lindsley et al, 2006; Chidiac and Angers, 2023). Consistent with this, in our protocol, the iPSCs are initially treated with FGF alone, while a Wnt agonist (CHIR99021, hereafter CHIRON) is added together with BMP2 at day 14 of differentiation (i.e., after 2–3 passages from the emergence of the first CNCCs). As aforementioned, BMP signalling is crucial to enable CNCC specification and differentiation (Mishina and Snider, 2014; Correia et al, 2007; Kanzler et al, 2000; Graf et al, 2016; Stuhlmiller and García-Castro, 2012; Roth et al, 2021; Liao et al, 2022; Bonilla-Claudio et al, 2012). In particular, a fine-tuned balance between Wnt and BMP pathways is essential for proper CNCC specification (Martik and Bronner, 2017; Simões-Costa and Bronner, 2015; Amel et al, 2023).

Based on this premise, we surmised that the mesodermal signature observed in the *CHD3*-KO cells might be caused by exposure to the Wnt agonist paired with the inability of the cells to properly respond to BMP2 stimuli, leading to a Wnt/BMP imbalance. Consistent with this, RNA-seq performed at day 14 of the CNCC specification protocol (i.e., right before exposure to CHIRON) revealed that the mesodermal markers are not yet expressed in the *CHD3*-KO cells before Wnt exposure (Fig. 4A). This finding supports the hypothesis that the primitive streak and mesodermal genes are eventually induced by the Wnt pulse.

Genome-wide analysis of the transcriptome at the day-14 timepoint identified 1411 differentially expressed genes (FDR < 5% and fold change > 1.5 or <−1.5; Dataset EV4; Appendix Fig. S7), with 569 being upregulated in the *CHD3*-KO and 842 downregulated (Fig. 4B). GO term analysis of the 842 downregulated genes revealed significant enrichment for BMP signalling response, morphogenesis, and patterning (Fig. 4C,D).

We next performed ATAC-seq at the same timepoint (day 14). Similar to day 18, at day 14, thousands of chromatin regions were either aberrantly open or aberrantly closed in the *CHD3*-KO cells (14,039 and 5493 regions, respectively, FDR < 5%; Fig. 4E; Appendix Fig. S8). Strikingly, the homeobox motif associated to the DLX/MSX BMP-responsive factors was the only motif enriched in the 5493 aberrantly closed regions at day 14 (i.e., day-14 *CHD3*-KO-specific peaks; Fig. 4F,G; regions whose accessibility was not affected by CHD3 loss were used as control for the differential motif analysis).

We further performed ATAC-seq timecourse (day 0, day 4 and day 10, in addition to day 14 and day 18) in order to establish whether CHD3 upregulation was necessary for the opening of these BMP-responsive regions. Notably, these experiments indicated that these regions are already accessible in iPSCs and during the neuroectodermal stage (when CHD3 expression is very low) in both *CHD3*-WT and *CHD3*-KO cells. However, these regions lose accessibility in the *CHD3*-KO cells only after day 10 (timepoint in which CHD3 protein is upregulated), once again mirroring CHD3 protein expression dynamics and suggesting that CHD3 might become responsible for the chromatin accessibility at these regions during CNCC induction and specification (Appendix Fig. S9).

Overall, the experiments conducted at day 14 suggest that CHD3 may have the role of priming the developing CNCCs to respond to BMP by maintaining chromatin accessibility at the BMP-responsive enhancers, facilitating the binding of homeodomain factors.

## CHD3 binds the BMP-responsive *cis*-regulatory elements

Next, we set out to investigate if CHD3 binds directly at the *cis*-regulatory elements that either lose or gain chromatin accessibility in the *CHD3*-KO cells, both at day 14 (before Wnt and BMP exposure) and day 18 (after Wnt and BMP exposure). We thus performed ChIP-seq for CHD3 at these two time points in *CHD3*-WT cells and detected CHD3 binding at nearly all the *CHD3*-WT-specific and *CHD3*-KO-specific ATAC-seq peaks at both time points (Fig. 5A–F), including at the *CHD3*-WT-specific sites containing BMP-responsive transcription factor motifs (Appendix Fig. S10). We also observed lack of CHD3 binding at ATAC-seq peaks conserved between *CHD3*-WT and *CHD3*-KO at these two time points (Appendix Fig. S10), suggesting CHD3 binding is exclusively observed at sites which gain or lose accessibility in the

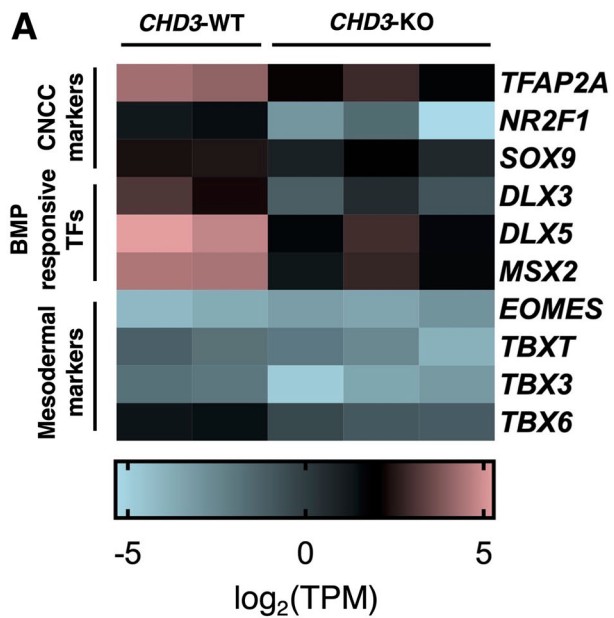

**A**

CHD3-WT    CHD3-KO

CNCC markers
- TFAP2A
- NR2F1
- SOX9

BMP responsive TFs
- DLX3
- DLX5
- MSX2

Mesodermal markers
- EOMES
- TBXT
- TBX3
- TBX6

log₂(TPM)
-5    0    5

**B**

## Differentially expressed genes in D14 CHD3-KO relative to CHD3-WT

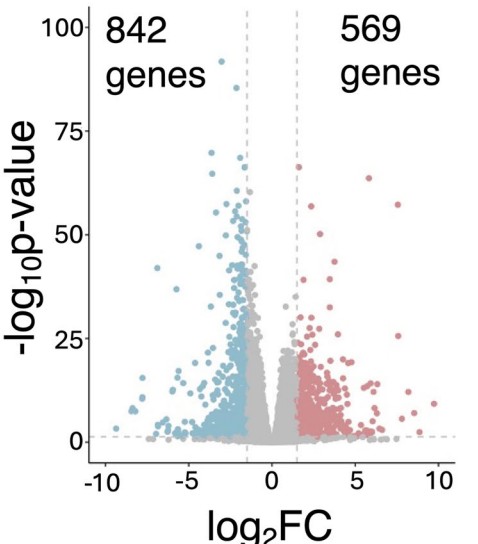

842 genes

569 genes

-log$_{10}$p-value

log$_{2}$FC

**C**

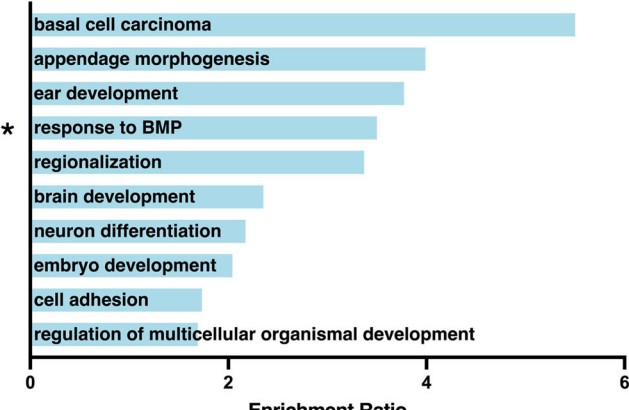

## Pathways enriched from genes downregulated in CHD3-KO D14 CNCCs

- basal cell carcinoma
- appendage morphogenesis
- ear development
- * response to BMP
- regionalization
- brain development
- neuron differentiation
- embryo development
- cell adhesion
- regulation of multicellular organismal development

Enrichment Ratio

**D**

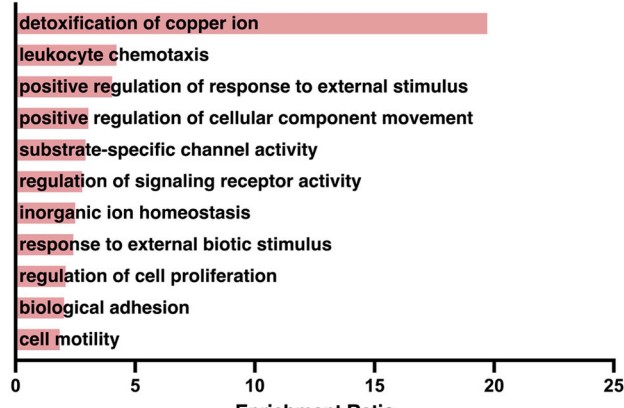

## Pathways enriched from genes upregulated in CHD3-KO D14 CNCCs

- detoxification of copper ion
- leukocyte chemotaxis
- positive regulation of response to external stimulus
- positive regulation of cellular component movement
- substrate-specific channel activity
- regulation of signaling receptor activity
- inorganic ion homeostasis
- response to external biotic stimulus
- regulation of cell proliferation
- biological adhesion
- cell motility

Enrichment Ratio

**E**

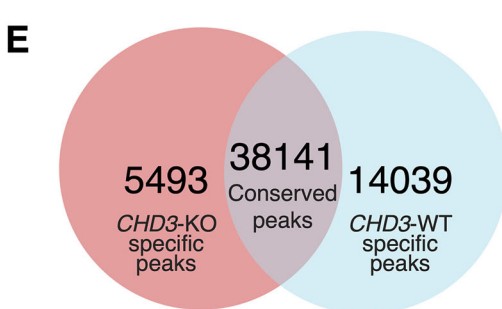

5493 CHD3-KO specific peaks

38141 Conserved peaks

14039 CHD3-WT specific peaks

**F**  *CHD3*-WT specific D14 ATAC-seq peaks

| Transcription Factor | Motif |
|---|---|
| MSX/DLX3 | |
| MSX/DLX5 | |
| LHX2 | |
| LHX1 | |
| MSX/DLX1 | |

**G**  *CHD3*-KO specific D14 ATAC-seq peaks

| Transcription Factor | Motif |
|---|---|
| JUN | |
| FOSL2 | |
| FRA2 | |
| ATF3 | |
| OCT4-SOX2-TCF-NANOG | |

◀ **Figure 4. The effects of CHD3 loss on gene expression and chromatin accessibility in D14 CNCCs.**

(A) Heatmap displaying expression of CNCC markers (*TFAP2A, NR2F1, SOX9*), genes involved in BMP signalling (*DLX3, DLX5, MSX2*) and mesodermal specification (*EOMES, TBXT, TBX3, TBX6*) in *CHD3*-WT (WT) and *CHD3*-KO (KO) day-14 CNCCs. (B) Volcano plot of differentially expressed genes in *CHD3*-KO relative to *CHD3*-WT in day-14 CNCCs ($n = 3$ biological replicates). Blue dots represent downregulated genes with $P$-adj $< 0.05$ and log$_2$FoldChange $< -1.5$. Red dots represent upregulated genes with $P$-adj $< 0.05$ and log$_2$FoldChange $> 1.5$. Adjusted $P$ values were calculated by DESeq2 (Love et al, 2014) using the Wald test. (C, D) GO term analysis of (C) significantly downregulated and (D) significantly upregulated genes in day-14 *CHD3*-KO CNCCs as determined using WebGestalt pathway analysis. (E) Venn diagram showing the number of *CHD3*-KO specific ATAC-seq peaks, *CHD3*-WT specific ATAC-seq peaks and conserved (peaks present in both *CHD3*-WT and *CHD3*-KO) ATAC-seq peaks in day-14 CNCCs. (F, G) Tables of motifs enriched in either (F) *CHD3*-KO-specific day-14 ATAC-seq peaks or (G) *CHD3*-WT-specific day-14 ATAC-seq peaks identified using HOMER. Source data are available online for this figure.

*CHD3*-KO, again supporting that CHD3 is required for chromatin accessibility at these sites. Further analysis revealed that the majority of CHD3-binding sites were either intronic (51% of day 14 and 44% of day 18 CHD3-binding sites) or intergenic (40% of day 14 and 41% of day 18 CHD3-binding sites), suggesting that CHD3 mainly binds putative distal *cis*-regulatory elements. In addition, using a publicly available H3K27ac ChIP-seq dataset generated in iPSC-derived CNCCs (Barnada et al, 2024b; Data ref: Barnada et al, 2024a), we found that approximately 30% of these intronic and intergenic CHD3-bound regions were decorated by H3K27ac, a marker of active enhancers (Appendix Fig. S10).

Given that most of the sites that are aberrantly closed at both day 14 and day 18 are enriched for the binding motif of the DLX/ MSX factors, we performed ChIP-seq for DLX5 at day 14 in *CHD3*-WT and *CHD3*-KO cells. We specifically selected DLX5 because it is highly expressed in our cell model at this timepoint and it has been previously associated with BMP response, CNCC specification and craniofacial development (Vu et al, 2021; Luo et al, 2001; Chung et al, 2010; Dash and Trainor, 2020). This experiment confirmed that DLX5 normally binds at these sites at day 14 and that loss of CHD3 results in depletion of DLX5 binding from these sites (Fig. 5G), possibly because of the significantly reduced chromatin accessibility, paired with downregulation of the *DLX5* gene.

In summary, our data indicate that CHD3 directly binds at important BMP-responsive *cis*-regulatory elements, and that loss of CHD3 binding from these elements attenuates their accessibility, ultimately affecting the ability of crucial transcription factors to bind. This leads to impairment in BMP response, which triggers an imbalance between signalling pathways.

## CHD3 acts both independently and with NuRD in CNCC specification

We next sought to establish whether CHD3 was acting as part of the NuRD complex to control accessibility at BMP-responsive sites during CNCC specification. To address this, we performed ChIP-seq for the core NuRD subunit MBD3 in day-14 *CHD3*-WT CNCCs and found 7171 peaks which were conserved across both biological replicates (FDR < 5%). Notably, the overlap between CHD3 peaks and MBD3 peaks was limited to ~10% of the MBD3 peaks and to ~10% of the CHD3 peaks, respectively (Appendix Fig. S11). It is worth noting that the CHD3 paralog CHD4 is highly expressed at this timepoint (*CHD4* median TPM: 195.8 vs *CHD3* median TPM: 19.4), suggesting that a CHD4-containing NuRD is likely also active at this timepoint. Cumulatively, these results might indicate that CHD3 could act both independently and as part of the NuRD complex during CNCC specification. However,

additional biochemical experiments will be required to support a NuRD-independent activity of CHD3.

## Titration of Wnt levels attenuates the aberrant mesodermal signature

Finally, we investigated whether attenuating Wnt signalling could rescue the aberrant phenotypes observed in *CHD3*-KO conditions. To achieve this, we differentiated *CHD3*-WT and *CHD3*-KO cells into CNCCs using different CHIRON concentrations: 3, 2 and 1 μM. At 3 μM (original protocol), the *CHD3*-KO cells displayed high expression of mesodermal markers both at gene and protein levels, paired with significantly reduced expression of CNCC markers (Fig. 6A,B). Notably, decreasing CHIRON (2 μM and 1 μM, respectively) was sufficient to significantly lower the activation of the mesodermal genes (e.g., *EOMES* and TBX3; Fig. 6A,B), but it was ineffective in restoring the expression of the CNCC markers (Fig. 6A,B).

Next, we tried to rescue the expression of the CNCC markers in *CHD3*-KO cells by simultaneously decreasing Wnt (1 μM of CHIRON) and increasing BMP (BMP2 concentration raised from 50 to 150 pg/ml). However, even subjecting the cells to three times more BMP2 was not effective in restoring the expression of the CNCC genes (Appendix Fig. S12), suggesting that the response to BMP is permanently impaired by the loss of CHD3.

Overall, these data support a model in which the aberrant primitive streak/early-mesoderm signature observed in *CHD3*-KO cells is caused by a Wnt/BMP imbalance which can be overcome by attenuating Wnt levels. On the other hand, the impaired CNCC specification is the result of ineffective BMP response, which cannot be rescued by simply increasing the amount of BMP ligand.

## Discussion

CHD3, a chromodomain helicase DNA-binding protein, is a core component of the NuRD complex, which modulates chromatin structure to regulate transcription (Bowen et al, 2004; Xue et al, 1998b). The NuRD complex is essential for developmental processes as it coordinates histone deacetylation and nucleosome remodelling to ensure precise gene expression patterns (Zhang et al, 2016; Denslow and Wade, 2007). CHD activity enables the NuRD complex to activate or repress gene expression through the increase of nucleosome density at gene promoters and enhancers (Bornelöv et al, 2018). The increased nucleosome density is thought to evict transcriptional machinery from the target sites, alter the chromatin landscape and enable the establishment of a new set of chromatin binding proteins which can either repress or promote transcription (Bornelöv et al, 2018).

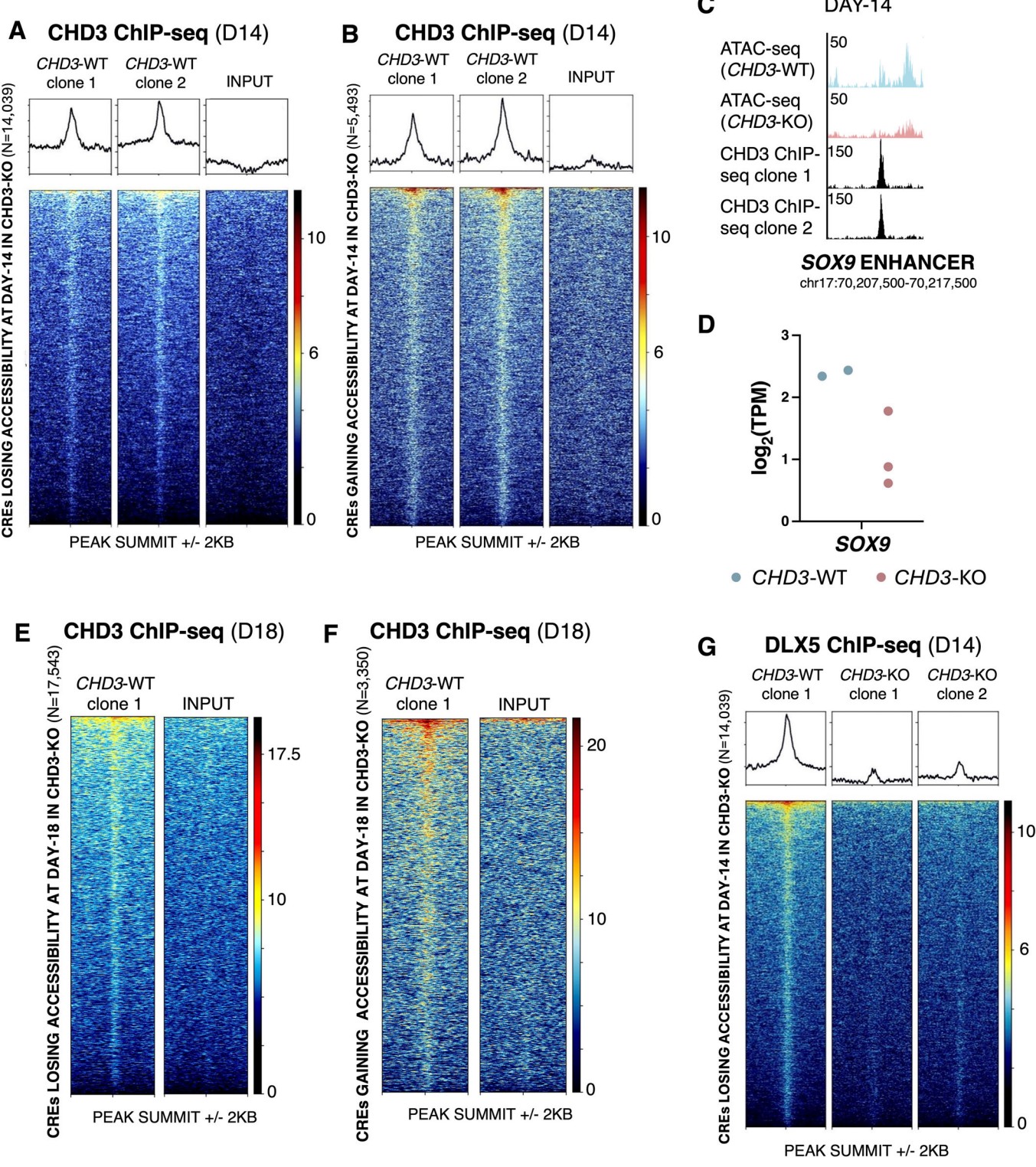

**Figure 5. CHD3 binding in CNCCs correlates with regions where chromatin accessibility is altered upon CHD3 loss.**

(**A, B**) Heatmap of CHD3 binding at *cis*-regulatory elements (CREs) which either (**A**) lose accessibility or (**B**) gain accessibility in day-14 *CHD3*-KO CNCCs. (**C**) An example of an enhancer (*SOX9* enhancer) bound by CHD3 in day-14 CNCCs which loses accessibility in *CHD3*-KO and (**D**) corresponding dot plot of log₂TPM for *SOX9* obtained from RNA-seq of day-14 *CHD3*-WT (*n* = 2 biological replicates) and *CHD3*-KO (*n* = 3 biological replicates) CNCCs. (**E, F**) Heatmap of CHD3 binding at *cis*-regulatory elements (CREs) which either (**E**) lose accessibility or (**F**) gain accessibility in day-18 *CHD3*-KO CNCCs. (**G**) Heatmap of DLX5 binding at *cis*-regulatory elements (CREs) which lose accessibility in day-14 *CHD3*-KO CNCCs. Source data are available online for this figure.

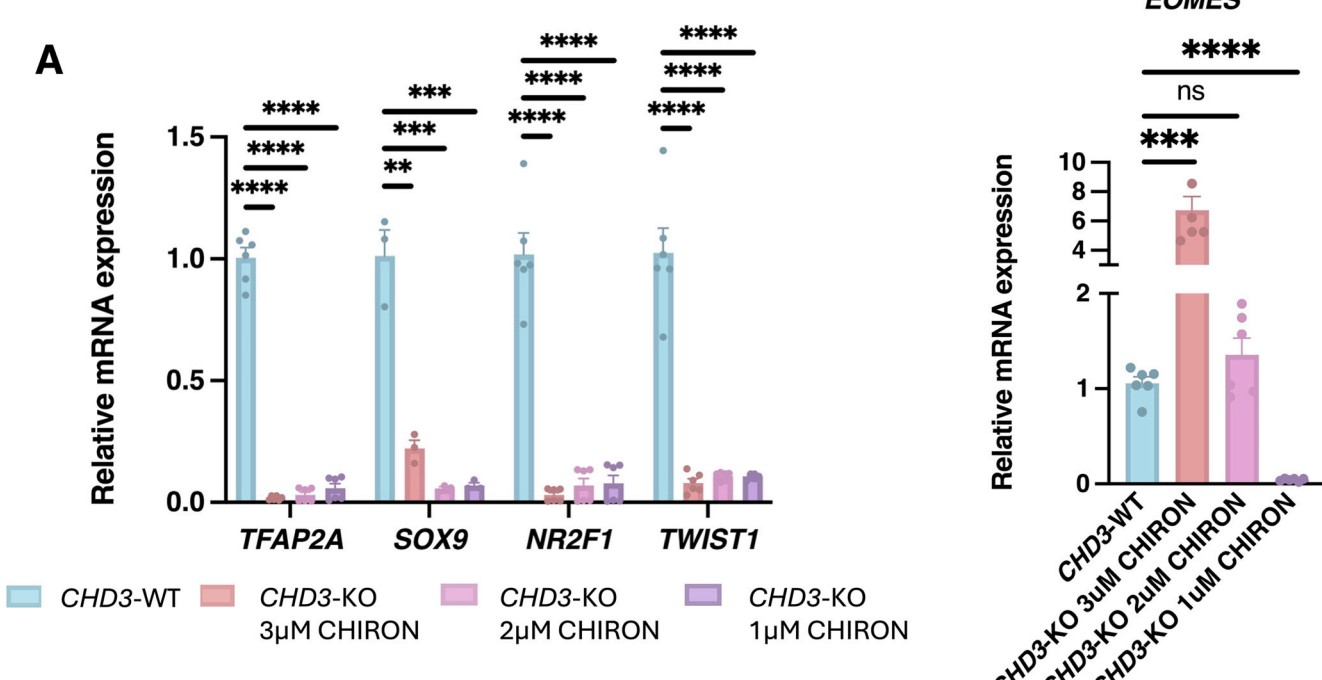

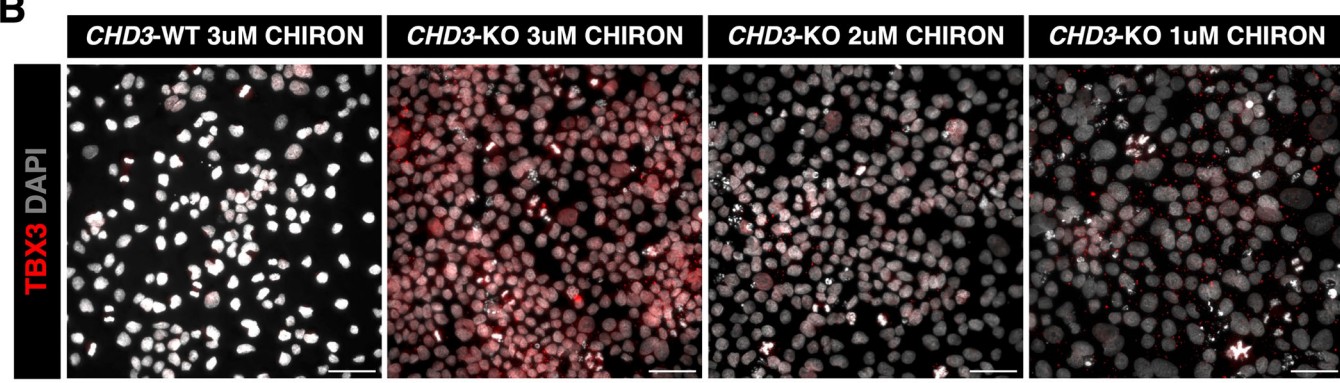

**Figure 6. CHD3-KO aberrant mesodermal phenotype is rescued by attenuated Wnt signalling.**

(A) RT-qPCR assessing the relative expression levels of CNCC markers (*TFAP2A, SOX9, NR2F1* and *TWIST*) and mesodermal marker (*EOMES*) between *CHD3*-WT and *CHD3*-KO day-18 CNCCs provided with either the standard 3 μM or a reduced concentration (2 μM or 1 μM) of CHIRON. Differences between conditions were assessed using unpaired Student's *t* test. \*\**P* < 0.01, \*\*\**P* < 0.001, \*\*\*\**P* < 0.0001, ns not significant (*n* = 6 with three technical replicates of two biological replicates for each condition). Data are presented as mean ± SEM. Exact *P* values: *TFAP2A*: *CHD3*-WT vs *CHD3*-KO 3 μM: *P* = <0.000001, *CHD3*-WT vs *CHD3*-KO 2 μM: *P* = <0.000001, *CHD3*-WT vs *CHD3*-KO 1 μM: <0.000001. *SOX9*: *CHD3*-WT vs *CHD3*-KO 3 μM: *P* = 0.002107, *CHD3*-WT vs *CHD3*-KO 2 μM: *P* = 0.000857, *CHD3*-WT vs *CHD3*-KO 1 μM: *P* = 0.000912. *NR2F1*: *CHD3*-WT vs *CHD3*-KO 3 μM: *P* = <0.000001, *CHD3*-WT vs *CHD3*-KO 2 μM: *P* = 0.000001, *CHD3*-WT vs *CHD3*-KO 1 μM: *P* = 0.000002. *TWIST1*: *CHD3*-WT vs *CHD3*-KO 3 μM: *P* = 0.000003, *CHD3*-WT vs *CHD3*-KO 2 μM: *P* = 0.000004, *CHD3*-WT vs *CHD3*-KO 1 μM: *P* = 0.000004. *EOMES*: *CHD3*-WT vs *CHD3*-KO 3 μM: *P* = 0.0001, *CHD3*-WT vs *CHD3*-KO 2 μM: *P* = 0.1438, *CHD3*-WT vs *CHD3*-KO 1 μM: *P* = <0.0001. (B) Immunofluorescence for the mesodermal marker TBX3 in *CHD3*-WT and *CHD3*-KO day-18 CNCCs provided with either the standard 3 μM or a reduced concentration (2 μM or 1 μM) of CHIRON. Scale bar: 50 μm. Source data are available online for this figure.

Heterozygous pathogenic variants in *CHD3* cause Snijders Blok–Campeau syndrome, a rare autosomal dominant neurodevelopmental syndrome characterised by a complex array of phenotypes that vary in severity between different affected individuals, including variable degrees of intellectual disability, impaired speech, macrocephaly, and distinct craniofacial features (Snijders Blok et al, 2018; Drivas et al, 2020). Many of the known pathogenic variants occur in the ATPase domain, which likely

compromises chromatin remodelling and disrupts developmental gene expression programmes, and loss-of-function alleles have also been reported (Snijders Blok et al, 2018; Drivas et al, 2020). The observations of distinctive craniofacial dysmorphisms in individuals with Snijders Blok–Campeau syndrome suggest that CHD3 plays a pivotal role in cranial neural crest cell (CNCC) development (Drivas et al, 2020; Snijders Blok et al, 2018). Although CHD3 has been implicated in neuronal migration and synapse formation

during brain development (Nitarska et al, 2016), its potential roles in CNCC specification and differentiation had never been investigated prior to the present study.

In this study, we demonstrate that CHD3 is indispensable for the proper specification of human CNCCs. Using CRISPR-Cas9-based knockout models, we showed that iPSCs with homozygous *CHD3* knockout retain pluripotent identity but exhibit severe defects in CNCC specification. Transcriptomic analysis revealed significant downregulation of CNCC markers such as *TFAP2A*, *SOX9*, and *TWIST1* in *CHD3*-KO cells, paired with upregulation of mesodermal markers like *TBXT*, *EOMES* and several others. These findings were corroborated by chromatin accessibility assays, which showed that a complete lack of CHD3 protein leads to the loss of open chromatin at cis-regulatory elements bound by BMP-responsive transcription factors, such as the DLX families, essential for CNCC specification (Luo et al, 2001; Zhu et al, 2023). Interestingly, mutations within *DLX5* and nearby enhancers have been found in patients with hearing loss and craniofacial anomalies similar to those seen in Snijders Blok–Campeau syndrome (Brown et al, 2010; Shamseldin et al, 2012; Birnbaum et al, 2012).

Our findings provide critical insights into the Wnt/BMP balance, a key regulatory axis during embryogenesis. BMP signalling is crucial for CNCC specification, acting through SMAD-dependent transcriptional programmes to induce patterning genes such as *MSX1* and *DLX5* (Mishina and Snider, 2014; Rahman et al, 2015; Roth et al, 2021). The specific role of BMP during the different stages of CNCC specification, migration and differentiation has been debated. Studies in *Xenopus* and zebrafish have suggested that BMP gradients are required to produce a specific level of BMP signalling which is permissive for neural crest formation (Marchant et al, 1998; Tríbulo et al, 2003; Nguyen et al, 1998). However, other studies in *Xenopus*, zebrafish, and chicks suggest that cranial neural crest induction is dependent on an initial inhibition of BMP signalling, followed by BMP activation (Patthey et al, 2009; Ragland and Raible, 2004; Steventon et al, 2009). In particular, BMP4 and BMP7 have been implicated in early cranial neural crest specification (Stuhlmiller and García-Castro, 2012). BMP signalling continues to play a key role in CNCC formation, enabling migration and later differentiation into derivative cells (Liao et al, 2022). Here, BMP2 appears to have an essential function in establishing migratory CNCCs (Correia et al, 2007; Kanzler et al, 2000), while BMP2, BMP4 and BMP7 are critical in enabling the subsequent formation of craniofacial structures (Bonilla-Claudio et al, 2012). In our study, *CHD3*-KO cells exhibited a marked reduction in BMP-responsive gene expression and chromatin accessibility, indicating that CHD3 is required for the transcriptional activation of BMP target genes.

Wnt signalling promotes mesodermal differentiation by stabilising beta-catenin, which activates mesoderm-specific transcription factors such as *EOMES*, *TBXT* and *GATA3* (Turner et al, 2014; Faial et al, 2015). In *CHD3*-KO cells, elevated Wnt signalling led to the aberrant upregulation of mesodermal markers, suggesting that CHD3 modulates the interplay between Wnt and BMP to ensure proper CNCC fate determination. FGF signalling, which synergizes with Wnt to promote mesodermal differentiation, also appeared to contribute to the aberrant mesodermal fate of *CHD3*-KO cells (Martik and Bronner, 2017). Notably, reducing Wnt levels attenuated the mesodermal signature, supporting the hypothesis that *CHD3*-deficient cells are unable to balance the Wnt signalling

in the absence of BMP response. However, this experiment failed to restore the expression of the CNCC markers, emphasising that CHD3 is indispensable for BMP signal transduction, independently from the relative contributions of the other signalling pathways.

This study establishes CHD3 as a pivotal chromatin remodeler that integrates BMP and Wnt signalling to regulate CNCC fate decisions. By modulating chromatin accessibility at BMP-responsive enhancers, CHD3 ensures the proper transcriptional activation of key developmental genes. Dysregulation of this balance, as seen in *CHD3*-KO cells, leads to anomalous mesodermal differentiation, providing a potential mechanistic explanation for the craniofacial defects observed in individuals with Snijders Blok–Campeau syndrome. We find that other NuRD-associated CHD paralogs (i.e., CHD4, CHD5) are not upregulated in *CHD3*-KO cells, suggesting that they may not be able to compensate for CHD3 loss in CNCCs, and that the BMP response during CNCC specification could be a CHD3-specific activity. How this tissue-specific activity of CHD3 is achieved is currently not understood and would need further investigation.

Importantly, the companion study where the CRISPR–iPSC lines were generated (preprint: den Hoed et al, 2024) investigated the role of CHD3 in cortical development and found that CHD3 is highly expressed in mature neurons, where it regulates synaptic development and function, suggesting that CHD3 may have different roles in different developmental processes.

Our work also suggests that the contributions of CHD3 to embryonic stem cell pluripotency are negligible, likely reflecting its relatively low expression levels in this cell type, where the paralog CHD4 has been demonstrated to be dominant and essential (Han et al, 2021; Hirota et al, 2019; Zhao et al, 2017).

In this study, we have also shown that the majority of CHD3-binding sites within CNCCs are not bound by the NuRD subunit MBD3, suggesting a potential function of CHD3 independent of the NuRD complex. CHD3 activity independent of NuRD has not previously been characterised, however it is known that the paralog CHD4 can act independently of NuRD, either as part of the ChAHP complex or alone (Ostapcuk et al, 2018; Amaya et al, 2013; Nitarska et al, 2016). CHD3 may therefore also act independently of NuRD in a similar manner, either as part of another complex or alone, however further studies are required in order to establish this. Moreover, since we only assessed MBD3 binding and the NuRD complex can contain either MBD2 or MBD3, we cannot rule out that CHD3 is preferentially incorporated into NuRD complexes containing MBD2 during CNCC specification. MBD2 and MBD3 are known to have distinct functions and target different regions of the genome (Schmolka et al, 2023; Menafra and Stunnenberg, 2014), therefore it is possible that the combination of MBD2 and CHD3 within the NuRD complex is needed during CNCC specification. This also requires further investigation to establish whether certain configurations of NuRD are more prominent within developing CNCCs. This may help shed further light on whether different NuRD subunits are preferentially combined at different time points or in different cells during development in order to enable distinct specification processes.

Future research should explore the interactions between CHD3-NuRD and other chromatin remodelling complexes, as well as its potential role in fine-tuning Wnt/FGF signalling during other steps of craniofacial development. In particular, given the importance of BMP in craniofacial osteogenesis, future studies should also explore

the function of CHD3 in the formation of CNCC-derived craniofacial bones and cartilage. In addition, in vivo studies using model organisms could further elucidate the developmental contexts in which CHD3 operates.

Finally, some limitations of our study should be acknowledged. First, this study was conducted using *CHD3* CRISPR-KO models and showed that *CHD3* heterozygous KO has no effect on CNCC specification, while *CHD3* homozygous KO has dramatic consequences on the same developmental process. However, it is important to highlight that a significant fraction of the individuals with Snijders Blok–Campeau syndrome present with heterozygous *CHD3* missense variants. We speculate that these heterozygous missense variants might have dominant negative effects, such that relevant disturbances in CNCC pathways can be captured in vitro by complete (homozygous) loss of CHD3, as in our system. Therefore, the present study would ideally be complemented by future research that employs cells from affected individuals carrying the specific heterozygous missense variants in the *CHD3* gene, or isogenic lines engineered to carry those same variants. Moreover, the "TAATTA" sequence, whose chromatin accessibility in CNCCs is regulated by CHD3, is recognised as a binding motif by most homeodomain factors, and not just DLX5. It is part of the CNCC "coordinator motif" (Prescott et al, 2015) and consequently we cannot exclude that other homeodomain factors (e.g., ALX family (Kim et al, 2024)) could also be implicated in CHD3-mediated BMP response.

## Methods

### Reagents and tools table

| Reagent/resource | Reference or source | Identifier or catalogue number |
|---|---|---|
| **Experimental models** | | |
| *CHD3*-WT clone 1 BIONi010-A | den Hoed et al (2024) (bioRxiv) | 010A$^{WT/WT}$C3 |
| *CHD3*-WT clone 2 BIONi010-A | den Hoed et al (2024) (bioRxiv) | 010A$^{WT/WT}$C2 |
| *CHD3*-HET-KO clone 1 BIONi010-A | den Hoed et al (2024) (bioRxiv) | 010A$^{WT/KO}$C3 |
| *CHD3*-HET-KO clone 2 BIONi010-A | den Hoed et al (2024) (bioRxiv) | 010A$^{WT/KO}$C2 |
| *CHD3*-KO clone 1 BIONi010-A | den Hoed et al (2024) (bioRxiv) | 010A$^{KO/KO}$C1 |
| *CHD3*-KO clone 2 BIONi010-A | den Hoed et al (2024) (bioRxiv) | 010A$^{KO/KO}$C2 |
| **Antibodies** | | |
| PE mouse anti-human TRA-1-60-R | BioLegend | 330609 |
| APC mouse anti-human SSEA-4 | BioLegend | 330417 |
| Rabbit anti-SOX2 | Active Motif | 39843 |
| Goat anti-NANOG | Bio-techne | AF1997 |
| Goat anti-OCT4 | Bio-techne | AF1759 |
| Mouse anti-TFAP2A | Fisher Scientific | 11594723 |
| Rabbit anti-CHD3 | Bethyl | A301-220A |

| Reagent/resource | Reference or source | Identifier or catalogue number |
|---|---|---|
| Goat anti-Brachyury | Bio-techne | AF2085 |
| Rabbit anti-TBX3 | Abcam | AB99302 |
| Donkey anti-Rabbit IgG (H + L) Highly Cross-Adsorbed Secondary Antibody, Alexa Fluor™ Plus 488 | Life Technologies | A32790 |
| Donkey anti-Goat IgG (H + L) Cross-Adsorbed Secondary Antibody, Alexa Fluor™ 594 | Life Technologies | A11058 |
| Goat anti-Mouse IgG2b Cross-Adsorbed Secondary Antibody, Alexa Fluor™ 647 | Life Technologies | A21242 |
| Goat anti-SOX17 | R&D Systems | 963121 |
| Goat anti-Otx2 | R&D Systems | 963273 |
| Goat anti-Brachyury | R&D Systems | 963427 |
| Rabbit anti-GAPDH | Cell Signaling Technology | 5174 |
| IRDye® 800CW Goat anti-Rabbit IgG | LI-COR | 926-32211 |
| Rabbit anti-DLX5 | Abcam | AB109737 |
| Rabbit anti-MBD3 | Abcam | AB157464 |
| **Oligonucleotides and other sequence-based reagents** | | |
| RT-qPCR primers | This study | Dataset EV5 |
| **Chemicals, enzymes and other reagents** | | |
| Geltrex | Thermo Fisher Scientific | A1413302 |
| mTeSR plus | Stem Cell Technologies | 100-0276 |
| Penicillin–streptomycin | Gibco | 15070063 |
| StemPro accutase | Gibco | A1110501 |
| Neurobasal medium | Gibco | 21103049 |
| DMEM/F12 | Gibco | A4192002 |
| B-27 supplement with vitamin A | Gibco | A3582801 |
| GlutaMAX supplement | Gibco | 35050061 |
| N-2 supplement | Gibco | 17502001 |
| Bovine insulin from pancreas | Sigma-Aldrich | I0516-5ML |
| EGF | Sigma-Aldrich | E9644-.2MG |
| FGF | Gibco | PHG0023 |
| Bovine serum albumin | Fisher Scientific | 12881630 |
| BMP2 | Gibco | 120-02-10UG |
| CHIRON 99021 | Stratech | S1263-SEL-5mg |
| PowerUp SYBR Green Master Mix for qPCR | Applied Biosystems | A25778 |
| Foetal bovine serum | Gibco | A4736201 |
| DAPI (4',6-Diamidino-2-Phenylindole, Dihydrochloride) | BioLegend | 422801 |
| Formaldehyde | Fisher Scientific | 10532955 |
| Triton X-100 | Merck | 648463 |
| Donkey serum | Abcam | AB7475 |
| Tween 20 | Promega | H5152 |

| Reagent/resource | Reference or source | Identifier or catalogue number |
|---|---|---|
| Fluorescence mounting medium | Agilent | S302380-2 |
| Radioimmunoprecipitation assay (RIPA) buffer | Thermo Scientific | 10230544 |
| Novex Tris-Glycine Mini Protein Gel | Invitrogen | XP04122BOX |
| Novex Tris-Glycine SDS running buffer | Invitrogen | LC26755 |
| Intercept (PBS) Blocking Buffer | LI-COR | 927-70001 |
| Disuccinimidyl Glutarate | Thermofisher | 20593 |
| Glycine | Thermo Scientific | 11454894 |
| Dynabeads Protein A | Invitrogen | 10002D |
| Proteinase K | Thermo Scientific | 10103533 |
| **Software** | | |
| FlowJo v10.9.0 | | https://www.flowjo.com |
| ImageJ | | https://imagej.net/ij/ |
| WebGestalt 2019 | | https://2019.webgestalt.org |
| R v4.2.2 | | https://www.r-project.org |
| GraphPad Prism v10.1.1 | | https://www.graphpad.com |
| UCSC Genome Browser | | https://genome.ucsc.edu |
| **Other** | | |
| Monarch Total RNA Miniprep Kit | NEB | T2010 |
| Maxima First Strand cDNA Synthesis Kit for RT-qPCR | Thermo Scientific | K1641 |
| Human Pluripotent Stem Cell Functional Identification Kit | R&D Systems | SC027B |
| Pierce BCA Protein Assay Kit | Thermo Scientific | 23225 |
| Wound Healing Assay Kit | Abcam | AB242285 |
| NEBNext Poly(A) mRNA Magnetic Isolation Module | NEB | E7490 |
| NEBNext Ultra II Directional RNA Library Prep Kit for Illumina | NEB | E7760 |
| ATAC-seq Kit | Active Motif | 53150 |
| ChIP DNA Clean & Concentrator Kit | Zymo | D5205 |
| Quantifluor ONE dsDNA System | Promega | E4871 |
| NEBNext Ultra II DNA Library Prep Kit for Illumina | NEB | E7645L |
| NEBNext Multiplex Oligos Dual Index Primers for Illumina | NEB | E7600S |

## Generation of the CRISPR–iPSC lines

Heterozygous and homozygous *CHD3* knockout iPSC lines were generated via CRISPR/Cas9 gene-editing in a companion paper

(preprint: den Hoed et al, 2024). Specifically, for this study, we used two different homozygous clones (*CHD3*-KO) in which Cas9 targeted the third exon of the *CHD3* gene in the established BIONi010-A iPSC line, producing a 1-base deletion (c.298delG) which generated a premature stop codon downstream (Fig. 1B) (preprint: den Hoed et al, 2024). The CRISPR/Cas9 strategy in this same BIONi010-A iPSC line also yielded heterozygous clones (*CHD3*-HET-KO) carrying c.298insA and c.298insT variants, respectively (preprint: den Hoed et al, 2024). All the lines were tested for mycoplasma and karyotyped using the KaryoStat HD Assay (preprint: den Hoed et al, 2024).

## Culturing of human iPSCs and differentiation into CNCCs

*CHD3*-WT, *CHD3*-HET-KO and *CHD3*-KO lines were expanded through culturing of the iPSCs on geltrex (Thermo Fisher Scientific, A1413302) coated wells in mTeSR plus medium (Stem Cell Technologies, 100-0276) containing 1% penicillin–streptomycin (Gibco, 15070063). The iPSCs were subsequently differentiated into CNCCs following the method developed by Prescott et al (Prescott et al, 2015). Briefly, cells were cultured in CNCC differentiation media (1:1 neurobasal medium and DMEM/F12, 1× penicillin/streptomycin, 0.5× B-27 supplement with vitamin A, 1× glutaMAX supplement, 0.5× N-2 supplement, 5 µg/ml bovine insulin from pancreas, 20 ng/ml EGF and 20 ng/ml FGF) for 6 days. On day 7, CNCC early maintenance media (1:1 neurobasal medium and DMEM/F12, 1× penicillin/streptomycin, 0.5× B-27 supplement with vitamin A, 1× glutaMAX supplement, 0.5× N-2 supplement, 1 mg/ml bovine serum albumin, 20 ng/ml EGF and 20 ng/ml FGF) was introduced and the cells were cultured in this up to day 14. On day 15, CNCC late maintenance media (1:1 neurobasal medium and DMEM/F12, 1× penicillin/streptomycin, 0.5× B-27 supplement with vitamin A, 1× glutaMAX supplement, 0.5× N-2 supplement, 1 mg/ml bovine serum albumin, 20 ng/ml EGF, 20 ng/ml FGF, 50 pg/ml BMP2 and 3 µM CHIRON 99021) was added, and cells were maintained in this up to day 18. Throughout differentiation, media was changed every other day and cells were passaged using StemPro Accutase (Gibco, A1110501) each time 80% confluency was reached.

## Reverse transcription quantitative polymerase chain reaction (RT-qPCR)

Cells were pelleted, and RNA was extracted using the Monarch Total RNA Miniprep Kit (NEB, T2010). In total, 600 ng of the extracted RNA was then converted to cDNA using the Thermo Scientific Maxima First Stand cDNA Synthesis Kit for RT-qPCR (Thermo Scientific, K1641). The qPCR reactions were prepared in a 96-well plate with each well containing 7.5 ng cDNA, 5 µl PowerUp SYBR Green Master Mix for qPCR (Applied Biosystems, A25778), 0.5 µM each of forward and reverse primers (Dataset EV5) and 1.5 µl of water for a total reaction volume of 10 µl. The qPCR was carried out using a Bio-rad Connect qPCR machine with the following conditions: 3 min at 95 °C, followed by 40 cycles of 10 s at 95 °C, 20 s at 63 °C and 30 s at 72 °C, with a final melt curve of 65 °C to 95 °C for 5 min. Technical and biological replicates were carried out for each sample, and 18S rRNA was used to normalise samples.

## Flow cytometry

Cells were treated with StemPro Accutase (Gibco, A1110501) for 5 min in order to produce a single-cell suspension. Cells were washed in cold PBS containing 2% foetal bovine serum (FBS) (Gibco, A4736201) and counted using a Countess automated cell counter. For each sample and timepoint, $1 \times 10^6$ cells were resuspended in 100 µl PBS-2% FBS and stained with 4 µl PE anti-human TRA-1-60-R Antibody (BioLegend, 330609), 2 µl APC anti-human SSEA-4 Antibody (BioLegend, 330417) and 500 ng/ml DAPI (4',6-Diamidino-2-Phenylindole, Dihydrochloride) (BioLegend, 422801). Cells were then incubated protected from light on ice for 15 min. Subsequently, cells were filtered into FACS tubes containing 300 µl of PBS-2% FBS, and flow cytometry was performed using the Agilent NovoCyte Penteon Flow Cytometer at the Sir Alexander Fleming building flow cytometry facility at Imperial College London. The resulting data were analysed using FlowJo software version 10.9.0.

## Immunofluorescence

Cells were plated onto geltrex coated coverslips and fixed using 4% formaldehyde (Fisher Scientific, 10532955) for 15 min at 37 °C. Samples were then permeabilised with 0.1% Triton X-100 (Merck, 648463) in PBS for 10 min at room temperature, before blocking for 1 h at room temperature in 10% donkey serum (Abcam, AB7475). Samples were incubated overnight at 4 °C on a rocker with primary antibodies of interest diluted in PBS. Negative controls with only PBS were also set up for each different secondary antibody. Samples were then washed in 0.1% Tween 20 (Promega, H5152) in PBS before being stained with the relevant secondary antibodies diluted in PBS for 1 h at 37 °C. Samples were washed again in 0.1% Tween 20 and stained with 1 µg/ml DAPI (BioLegend, 422801) for 15 min at room temperature. Coverslips were mounted using fluorescence mounting medium (Agilent, S302380-2). Stained cells were visualised using the Zeiss Axio Observer inverted microscope in the Facility for Imaging by Light Microscopy (FILM) at Imperial College London. The primary antibodies used were: anti-SOX2 (rabbit, 1:500, Active Motif 39843), anti-NANOG (goat, 1:50, Bio-techne AF1997), anti-OCT4 (goat, 1:50, Bio-techne AF1759), anti-TFAP2A (mouse, 1:100, Fisher Scientific 11594723), anti-CHD3 (rabbit, 1:500, Bethyl A301-220A), anti-Brachyury (goat, 1:20, Bio-techne AF2085) and anti-TBX3 (rabbit, 1:120, Abcam AB99302). The secondary antibodies used were: Donkey anti-Rabbit IgG (H + L) Highly Cross-Adsorbed Secondary Antibody, Alexa Fluor™ Plus 488 (1:500, Life Technologies, A32790), Donkey anti-Goat IgG (H + L) Cross-Adsorbed Secondary Antibody, Alexa Fluor™ 594 (1:500, Life Technologies, A11058) and Goat anti-Mouse IgG2b Cross-Adsorbed Secondary Antibody, Alexa Fluor™ 647 (1:500, Life Technologies, A21242).

## Trilineage differentiation

iPSCs were differentiated into the three germ layers (endoderm, ectoderm and mesoderm) using the Human Pluripotent Stem Cell Functional Identification Kit (R&D Systems, SC027B). Expression of relevant markers was then assessed using immunofluorescence with 10 µg/ml of the antibodies provided in the kit (Goat anti-human SOX17, Goat anti-human Otx2 and Goat anti-human

Brachyury) and Donkey anti-Goat IgG (H + L) Cross-Adsorbed Secondary Antibody, Alexa Fluor™ 594 (1:500).

## Western blot

For the western blot, cells were harvested and washed three times in 1× phosphate-buffered saline (PBS) and pelleted. Protein was extracted using radioimmunoprecipitation assay (RIPA) buffer (Thermo Scientific, 10230544) with protease inhibitor (PI) and was quantified using the Pierce BCA Protein Assay Kit (Thermo Scientific, 23225). In total, 30 µg of each protein sample were loaded onto a Novex Tris-Glycine Mini Protein Gel (Invitrogen, XP04122BOX). Proteins were separated using gel electrophoresis in 1× SDS running buffer and were then transferred to a nitrocellulose membrane. The membrane was blocked in Intercept (PBS) Blocking Buffer (LI-COR, 927-70001) for 1 h. Primary antibodies, anti-CHD3 (rabbit, Bethyl A301-220A) and anti-GAPDH (rabbit, Cell Signaling Technology 5174), were diluted 1:1000 in Intercept (PBS) Blocking Buffer containing 0.2% Tween 20. The membrane was incubated in the primary antibody dilution on rollers overnight at 4 °C. The membrane was then washed four times for 5 min in 1× PBS containing 0.1% Tween 20 (PBST) on a rocker. The secondary antibody IRDye® 800CW anti-Rabbit IgG (goat, LI-COR 926-32211) was diluted 1:15,000 in Intercept (PBS) Blocking Buffer containing 0.2% Tween 20. The membrane was incubated in the secondary antibody protected from light at room temperature for 1 h on a rocker. The membrane was then washed again in PBST four times for 5 min on a rocker before being imaged using the LI-COR Odyssey XF imaging system.

## Scratch wound assay

CNCCs were cultured as described until day 16. On day 16, cells were harvested using StemPro Accutase (Gibco, A1110501) and counted. In total, 500,000 cells were seeded onto each well of a geltrex (Thermo Fisher Scientific, A1413302) coated 24-well wound-healing assay plate containing a plastic insert (Abcam, AB242285) which generated a 0.9 mm wound. Twelve wells were set up for each clone (*CHD3*-WT clone 1, *CHD3*-WT clone 2, *CHD3*-KO clone 1, *CHD3*-KO clone 2). Growth factors (EGF and FGF) were then removed from the CNCC late maintenance media for 24 h prior to the start of the assay to prevent proliferation. On day 18, the inserts were removed and wells were washed twice with PBS before fresh CNCC late maintenance media (-EGF, -FGF) was added. Images of each well were taken using a light microscope at 0, 6 and 24 h after the removal of the inserts. The percentage of wound closure was then calculated for each well at 6 h and 24 h using the wound-healing size tool plugin (Suarez-Arnedo et al, 2020).

## RNA-sequencing

Cells were pelleted, and RNA extraction was performed using the Monarch Total RNA Miniprep Kit (NEB, T2010). RNA was quantified with a nanodrop, and the quality was assessed using TapeStation 2200 (Agilent Technologies). Only RNA with a RIN score above 8 was used. RNA libraries were prepared from 1 µg of RNA input using the NEBNext Poly(A) mRNA Magnetic Isolation Module (NEB, E7490) and NEBNext Ultra II Directional RNA

Library Prep Kit for Illumina (NEB, E7760). Sequencing was carried out by the Wistar Institute to generate 60 bp paired-end reads. Two replicates of clone 1 and one replicate of clone 2 were used at each specified timepoint for all of the three conditions (*CHD3*-WT, *CHD3*-HET-KO and *CHD3*-KO). This corresponded to two biological replicates and a total of three technical replicates per condition.

## RNA-sequencing analysis

First, adapters were removed using TrimGalore!, then reads were mapped and quantified using Kallisto (Bray et al, 2016). Differential gene expression was analysed using DESeq2 (Love et al, 2014). Gene set enrichment analysis was performed using WebGestalt 2019 (Liao et al, 2019). Additional statistical analysis was carried out using R (version 4.2.2) and GraphPad Prism (version 10.1.1).

## ATAC-sequencing

ATAC-seq was performed using 50,000 cells per sample. Libraries were prepared using the ATAC-seq Kit (Active Motif, 53150) following the manufacturer's instructions. Sequencing was carried out by the Wistar Institute to generate 60 bp paired-end reads. One replicate of clone 1 and one replicate of clone 2 were used at each specified timepoint for both conditions (*CHD3*-WT and *CHD3*-KO). This corresponded to two biological replicates per condition.

## ATAC-sequencing analysis

Adapters were removed using TrimGalore! and the reads were then aligned to the hg19 human reference genome using the Burrows–Wheeler Alignment (BWA) tool with the MEM algorithm (Li and Durbin, 2009). SAMTools (Li et al, 2009) was then used to filter for high-quality (MAPQ > 10) reads and to remove PCR duplicates. Peaks were then called using MACS2 (Zhang et al, 2008) with 5% FDR. Consensus peaks (peaks found in all replicates) were identified for each cell line using BEDTools (Quinlan and Hall, 2010) and these were used for subsequent analyses. ATAC-seq peaks were visualised using UCSC Genome Browser (Nassar et al, 2023). Motif analysis was performed using HOMER (Heinz et al, 2010). All further downstream analysis was performed using BEDTools (Quinlan and Hall, 2010) and deepTools (Ramírez et al, 2014).

## ChIP-sequencing

Two replicates were performed for each condition. For each replicate, 11 million cells were cross-linked using 1% formaldehyde for 5 min at room temperature. For MBD3 ChIP, cells were double cross-linked using 2 mM disuccinimidyl glutarate (DSG) for 20 min followed by 1% formaldehyde for 10 min at room temperature. Cells were then quenched with 125 mM glycine for 5 min at room temperature before being washed twice with 1× PBS. The fixed cells were then resuspended in ChIP buffer (150 mM NaCl, 1% Triton X-100, 5 mM EDTA, 10 mM Tris-HCl, 0.5 mM DTT, 0.3% SDS, protease inhibitor) and incubated on ice for 10 min. Chromatin was sheared to an average length of 100–1000 bp using a Covaris M220 Focused-Ultrasonicator at 5% duty factor for 7-27 min. The

chromatin lysate was diluted in SDS-free ChIP buffer. In total, 10 μg of antibody was used for both CHD3 (Bethyl Laboratories, A301-220A) and DLX5 (Abcam, AB109737), and 5 μg was used for MBD3 (Abcam, AB157464). The antibody was added to at least 5 μg of sonicated chromatin along with Dynabeads Protein A (Invitrogen, 10002D) and incubated overnight at 4 °C with rotation. The beads were then washed twice with each of the following buffers: Mixed Micelle Buffer (150 mM NaCl, 1% Triton X-100, 0.2% SDS, 20 mM Tris-HCl, 5 mM EDTA, 65% sucrose), Buffer 200 (200 mM NaCl, 1% Triton X-100, 0.1% sodium deoxycholate, 25 mM HEPES, 1 mM EDTA), LiCl detergent wash (250 mM LiCl, 0.5% sodium deoxycholate, 0.5% NP-40, 10 mM Tris-HCl, 1 mM EDTA) and a final wash was performed with cold 0.1× TE. Finally, beads were resuspended in 1× TE containing 1% SDS and incubated at 65 °C for 10 min to elute immunocomplexes. The elution was repeated twice, and the samples were incubated overnight at 65 °C to reverse cross-linking, along with the input (5% of the starting material). The DNA was digested with 0.5 mg/ml Proteinase K for 1 h at 65 °C and then purified using the ChIP DNA Clean & Concentrator kit (Zymo, D5205) and quantified with the QuantiFluor ONE dsDNA system (Promega, E4871). Barcoded libraries were made with NEBNext Ultra II DNA Library Prep Kit for Illumina (NEB, E7645L) using NEBNext Multiplex Oligos Dual Index Primers for Illumina (NEB, E7600S). Sequencing was carried out by the Wistar Institute to generate 60 bp paired-end reads or by Novogene to generate 150 bp paired-end reads. Clones 1 and 2 were used as biological replicates.

## ChIP-sequencing analysis

Adapters were removed with TrimGalore! and the sequences were aligned to the reference genome hg19, using Burrows–Wheeler Alignment tool, with the MEM algorithm (Li and Durbin, 2009). Uniquely mapping aligned reads were filtered based on mapping quality (MAPQ > 10) to restrict our analysis to higher quality and uniquely mapped reads, and PCR duplicates were removed. HOMER (Heinz et al, 2010) was used to call peaks using the default parameters at 5% FDR. All statistical analyses were performed using BEDTools (Quinlan and Hall, 2010), deepTools (Ramírez et al, 2014), R (version 4.2.2) and GraphPad Prism (version 10.1.1).

# Data availability

Further information and requests for resources and reagents should be directed to and will be fulfilled by the lead contact, Marco Trizzino (m.trizzino@imperial.ac.uk). Inquiries on CRISPR cell lines used in this study should be directed to Prof. Simon E Fisher. RNA-seq, ATAC-seq, and ChIP-seq data have been deposited in the Gene Expression Omnibus (GEO) under accession code GEO: GSE288669 and are publicly available as of the date of manuscript submission.

The source data of this paper are collected in the following database record: biostudies:S-SCDT-10_1038-S44319-025-00555-w.

# Peer review information

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

## Acknowledgements

We thank Dr. Samantha Barnada (Thomas Jefferson University) and Prof. Brian Hendrich (University of Cambridge) for insightful discussions on the data. The authors thank the Genomic Facility at The Wistar Institute (Philadelphia, PA) for the Next Generation Sequencing, and the FACS and Imaging Facilities at Imperial College London. For this work, MT was funded by the G Harold and Leila Y Mathers Foundation and by Biotechnology and Biological Sciences Research Council (BBSRC, grant BB/Y000854/1).

## Author contributions

**Zoe H Mitchell**: Data curation; Formal analysis; Investigation; Methodology; Writing—original draft. **Joery den Hoed**: Resources; Writing—review and editing. **Willemijn Claassen**: Resources; Writing—review and editing. **Martina Demurtas**: Data curation; Investigation; Writing—review and editing. **Laura Deelen**: Data curation; Investigation; Writing—review and editing. **Philippe M Campeau**: Conceptualisation; Resources; Writing—review and editing. **Karen Liu**: Conceptualisation; Investigation; Methodology; Writing—review and editing. **Simon E Fisher**: Resources; Supervision; Writing—review and editing. **Marco Trizzino**: Conceptualisation; Resources; Data curation; Formal analysis; Supervision; Funding acquisition; Investigation; Writing—original draft; Project administration.

Source data underlying figure panels in this paper may have individual authorship assigned. Where available, figure panel/source data authorship is listed in the following database record: biostudies:S-SCDT-10_1038-S44319-025-00555-w.

## Disclosure and competing interests statement

The authors declare no competing interests.

