## [Peer Review File · EMBO Reports]

The NuRD component CHD3 promotes BMP signalling during cranial neural crest cell specification

Zoe Mitchell, Joery den Hoed, Willemijn Claassen, Martina Demurtas, Laura Deelen, Philippe Campeau, Karen Liu, Simon Fisher, and MARCO TRIZZINO

Corresponding author(s): MARCO TRIZZINO (m.trizzino@imperial.ac.uk)

Review Timeline:

Transfer Date:	6th May 25
Editorial Decision:	12th May 25
Revision Received:	3rd Jun 25
Editorial Decision:	8th Jul 25
Revision Received:	11th Jul 25
Accepted:	21st Jul 25

**Transaction Report: This manuscript was transferred to
EMBO reports following peer review at Review Commons.**

**Review
COMMONS**

Review #1

1. Evidence, reproducibility and clarity:

Evidence, reproducibility and clarity (Required)

In this manuscript, Mitchell et al examine the impact of CHD3 KO (or het) on iPSC differentiation to CNCCs to model how pathogenic CHD3 mutations promote a neurodevelopmental disorder. The authors perform genomic characterization of the KO and het mutants during this differentiation model, and identify loss of CHD3 results in BMP target gene repression and increased mesoderm fate. Finally, the authors attempt to rescue the observed defects by titrating Wnt levels and observe partial rescue. I think the paper is easy to follow, contains interesting data, and establishes a novel role for CHD3 in CNCC differentiation, which may have implications in the disorder highlighted. I have the following suggestions:

1. Figure 1 presents nice confirmation of the CHD3 KO cell lines being used. However, given that these cell lines were previously published, I suggest moving these data to the supplement.
2. In the results section for Figure 1, the authors discuss the CHD3 heterozygotes, but I only see the KO cell line data presented. It would be especially nice to see the protein levels of Chd3 in the het.
3. The authors discuss which genes are up and downregulated in the Chd3 KO D18 RNAseq, and show a clear heatmap in Figure 2A for WT cells. The same heatmap for candidate genes discussed in the results would be appreciated for Chd3 KO. Only a subset of markers are shown in Fig 2C.
4. In general 2-3 replicates are presented. While the authors are showing heatmaps for selected locations for individual clones, which is appreciated (ex: Figure 4B and Fig 6), the QC for data quality is missing. For example, show spearman correlation across the genome for datasets as a supplement.
5. In the section discussing the results presented in Figure 4, the authors discuss the ATAC-seq peak number changes and overlap with gene expression changes. However, the overlap with gene expression changes is not shown. Making a simple venn diagram would help readers.
 - a. In addition, showing a heatmap for unchanged ATACseq peaks can help to demonstrate the increase/decrease.
6. In Figure 6, the authors present ChIPseq data for CHD3 in D14 and D18 samples, focusing on locations losing or gaining accessibility. What is enrichment at unchanged sites? Is CHD3 specifically enriched at changed locations? Then what about over genes

with altered gene expression vs not changed? Is CHD3 only bound to distal elements? Performing an analysis of the peak distribution, perhaps with ChromHMM or other methods to look at promoter vs enhancer vs other locations. These types of analyses could really enrich the interpretation of direct CHD3 function.

7. Given the changes in the CHD3 KO accessibility are mostly gene distal, are there existing Hi-C/microC/promoter CaptureC or other that can be used to ask if these are interacting with the predicted genes?

8. Are the results observed NuRD-based or CHD3 NuRD independent functions? Looking at other NuRD subunit binding or effects in differentiation would help to dig into this a bit more. I realize this is a bit of a big ask, so I am not asking for everything. Are there existing binding data in CNCCs for a NuRD subunit that could be examined for overlap in where these changes occur, for example? I want to be clear I am not asking the authors to do all the experiments for an alternative NuRD subunit.

9. The authors observe defects in CNCCs through genomic experiments. It would be really nice to perform simple wound healing/scratch assays and/or transwell assays to test if the CNCC migration phenotype is reduced in the CHD3 KO as well which would support the transcriptomic data.

10. Related to the above, I am not sure if there is a phenotypic test for enhanced mesoderm. I suspect only IF/expression and morphology are possible, which the authors did. However, sorting the cells (with some defined markers) to ask how many are mesoderm-like vs CNCC in WT vs CHD3 KO would give some information outside of the bulk expression data.

11. I did not see a reviewer token for the GEO data, so I could not check the deposited datasets.

****Minor points****

12. 1A seems to fit better with Figure 2.

13. The authors say that the KO cell lines are not defective in pluripotency, but Figures 1G suggests a slight decrease in SSEA-1. Is this reproducibly observed?

14. Would be nice to show number of up and downregulated genes in volcano plots for fast viewing of readers (ex: Fig 2B).

15. Is it fair to use violin plots when data points are only 2-3 replicates (as in Figures 2C, 3D)

16. The labels in Fig 4A and 5E are very hard to read.

17. For browser tracks, the authors show very zoomed in examples (Fig 4C, and especially Fig 6C). showing a bit more of the area around these peaks would give readers a more clear appreciation of the data.

18. Related to browser tracks, including more information just as including the gene

expression changes (such as in Fig 6C) to enhance the interpretation of the impact of Chd3 binding, accessibility change and then, I presume, reduced Sox9 expression. Similar suggestion for Figure 4C, where I anticipate coordinate transcription changes of the associated genes.

19. Do the authors observe any clone variability between the two CHD3 KO clones? There is variability I see in some of the heatmaps, but don't know if that it is because of clones or technical variation.

****Referees cross-commenting****

I think that the other reviewer and I are inline with each other in terms of our reviews and thoughts on the manuscript, so I do not have anything to add.

2. Significance:

Significance (Required)

The paper presented by Mitchell et al represents a new role for CHD3 in regulating CNCC differentiation and perhaps explains why CHD3 mutations exist in neurodevelopmental disorders such as Snijders Blok-Campeau Syndrome. Limitations are the reliance on genomic datasets and modeled differentiation, although this permits for more mechanistic studies.

I believe the fields of neural development, stem cell, chromatin biology, and others will be interested in this manuscript.

3. How much time do you estimate the authors will need to complete the suggested revisions:

Estimated time to Complete Revisions (Required)

(Decision Recommendation)

Between 3 and 6 months

4. Review Commons values the work of reviewers and encourages them to get credit for their work. Select 'Yes' below to register your reviewing activity at Web of Science Reviewer Recognition Service (formerly Publons); note that the content of your review will not be visible on Web of Science.

No

Review #2

1. Evidence, reproducibility and clarity:

Evidence, reproducibility and clarity (Required)

Summary

In this manuscript, Mitchell et al. study the function of CHD3 - a member of the NuRD chromatin remodeling complex - during human cranial neural crest cells (CNCC) differentiation in vitro. To this end they use iPSC CHD3-KO lines. They first observed this deletion has no impact on pluripotency levels of mutant iPSC neither on their capacity to differentiate into the three germ layers derivatives. Yet, they showed these mutant cells are unable to form CNCC as they fail to induce EMT genes and undergo CNCC differentiation. Using ATACseq, the authors then showed CHD3 KO present a reorganization of the chromatin accessible landscape, biasing these cells from a CNCC to a mesoderm fate. They further determine that upon differentiation of CHD3 KO cells, BMP responsive regulatory elements are aberrantly closed, making the cells insensitive to the signaling, explaining how they then fail to generate CNCC. Using ChIP-seq, they confirmed a direct action of CHD3 in making these elements accessible as it normally binds to these chromatin regions to allow proper differentiation. In addition, they demonstrate these BMP responsive genes are bound by DLX5, a transcription factor essential for neural crest development. Finally, the authors showed that during CNCC differentiation, CHD3 KO cells experience an imbalance between BMP and WNT signaling, leading to these cells adopting a mesoderm instead of a CNCC identity. They finally, showed this can be partially rescued by reducing the amount of Wnt signaling - that decreases the mesoderm gene expression - however, it not sufficient to induce a neural crest identity.

Major comments

1. What is the expression level of CHD3 in the heterozygote line? Does the remaining allele compensate for the loss which will explain the absence of phenotype?
2. Since CHD3 shows a progressive upregulation in expression during CNCC differentiation (Fig. 2E), one hypothesis can be that it is not necessary involved in the activation of the CNCC programs but instead it is involved in maintaining these programs active - by keeping regulatory elements accessible. Thus, authors should check expression of CNCC markers,

and EMT genes at the same time point than Fig. 2E in both WT and KO cells.

3. The authors should use the term "regulatory elements" instead of "enhancers" as they can act either as activator or repressors.

4. On the same line, while the authors indicate "Motif analysis of the enhancers aberrantly active in CHD3-KO cells ", they haven't shown these are active. They should say they perform the analysis on regulatory elements aberrantly accessible in CHD3 KO.

5. The rationale that led the authors to focus on genes typically expressed in the primitive streak and in the early pre-migratory mesoderm, and BMP responsive transcription factors could be better explained. Are they part of the most deregulated genes in the RNAseq analysis?

6. In the absence of CHD3, BMP response is not effective. While the authors nicely showed this is linked with changes in chromatin accessibility, it is necessary to check the expression levels of BMP receptors in CHD3 KO cells.

7. Aberrant early mesoderm signature of the CHD3-KO cells needs to be better shown. It is not obvious from the GO analysis in Fig. 2 and the authors then showed expression of some markers but it is unclear how they picked them up.

8. It has been shown that CNCC regulatory elements controlling differentiation genes are primed/accessible prior migration (PMID: 31792380; PMID: 33542111). Since the authors claim "CHD3 may have the role of priming the developing CNCCs to respond to BMP by opening the chromatin at the BMP responsive enhancers", it will be good to perform ATACseq at several time points during the differentiation process to assess the dynamic of chromatin reorganization to see when the switch to mesoderm fate occurs and how accessibility of BMP responsive element changes in WT and KO cells during CNCC differentiation to be able to demonstrate the KO fail to make BMP responsive element accessible or whether it is a defect in the maintenance of this accessibility.

9. The authors claim CHD3 directly binds at BMP responsive enhancers, but in the figure, they show the data for all the region gaining or losing activity. It will be nice to add the information for the BMP responsive elements only.

10. Motifs enrichment analysis of regions gaining accessibility in CHD3 KO do not seem to be labeled as Wnt responsive elements. The authors need to support better that CHD3 KO express more Wnt signaling/activity.

11. OPTIONAL: Does increasing BMP concentration early during CHD3 KO differentiation has a better effect at rescuing CNCC differentiation?

12. OPTIONAL: While this is not necessary for the current study, it is very intriguing that other CHD family member do not compensate. How this tissue or DNA sequence activity is achieved could be discussed. What are CHD4 or CHD5 expressed during CNCC differentiation? Could they be used to rescue the CHD3 KO phenotype? While this may be difficult to test, it could perhaps be discussed.

****Minor comments****

13. In the discussion, the authors could indicate whether CHD3 mutants somehow phenocopies some of the craniofacial defects observed in DLX5 mutant patients.

14. It is not indicated were to find the data regarding expression epithelial and mesenchymal genes in the CHD3-KO cells.

15. Authors could add in the discussion what is known about how CHD3 function changes from opening or closing chromatin is very intriguing a could be discussed.

2. Significance:

Significance (Required)

General assessment:

The link between chromatin remodelers and craniofacial defects has been shown in several studies in the past, but it still remains unclear how mutation of a given factor leads to such tissue specific defects. This manuscript represents an interesting and detailed mechanistic study on the role of chromatin remodeler in cell fate decision, showing that reorganization of chromatin accessibility is essential to proper response to signaling pathway and cell differentiation.

Advance:

The authors manage to link how mutant-induced changes in chromatin accessibility biased the cells towards a mesoderm fate as they directly impact the capacity of the cells to respond to BMP signaling - these regions closing upon CHD3 loss. However, the question remains to know whether CHD3 acts as an initiating factor or instead is involved in maintaining these programs active. As noted by the authors, a clinical link (with patient-derived iPCS) would be of great interest but as it stands the story already provide a good mechanistic understanding on how CHD3 control CNCC differentiation.

Audience:

This manuscript will be of great interest for specialized audience, yet a broader public may find it interesting too.

Reviewer field of expertise:

Neural crest and craniofacial development, epigenetics, transcriptomics

3. How much time do you estimate the authors will need to complete the suggested revisions:

Estimated time to Complete Revisions (Required)

(Decision Recommendation)

Between 3 and 6 months

Yes

Dear Marco,

Thank you for the submission of your research manuscript to EMBO Reports. As discussed, we would like to invite you to revise your study according to your revision plan.

We realize that it is difficult to revise to a specific deadline. In the interest of protecting the conceptual advance provided by the work, we recommend a revision within 3 months (August 12th). Please discuss the revision progress ahead of this time with the editor if you require more time to complete the revisions.

I am also happy to discuss the revision further via e-mail or a video call, if you wish.

Please find below the formatting instructions for EMBO Reports. I want to draw your attention to point (7), the Data availability section. Please note that we need an URL that resolves directly to the datasets at GEO. Please remove references to "data available upon request". This section should only refer to the deposited data.

The Supplementary Files S1-S4 should be relabeled to Dataset EV# and they need a legend in a separate tab.

2) individual production quality figure files as .eps, .tif, .jpg (one file per figure). Please download our Figure Preparation Guidelines (figure preparation pdf) from our Author Guidelines pages <https://www.embopress.org/page/journal/14693178/authorguide> for more info on how to prepare your figures.

4) a complete author checklist, which you can download from our author guidelines (). Please insert information in the checklist that is also reflected in the manuscript. The completed author checklist will also be part of the RPF.

5) Please note that all corresponding authors are required to supply an ORCID ID for their name upon submission of a revised manuscript (). Please find instructions on how to link your ORCID ID to your account in our manuscript tracking system in our Author guidelines ()

6) We replaced Supplementary Information with Expanded View (EV) Figures and Tables that are collapsible/expandable online. A maximum of 5 EV Figures can be typeset. EV Figures should be cited as "Figure EV1, Figure EV2" etc... in the text and their respective legends should be included in the main text after the legends of regular figures.

7) Before submitting your revision, primary datasets (and computer code, where appropriate) produced in this study need to be deposited in an appropriate public database (see < <https://www.embopress.org/page/journal/14693178/authorguide#dataavailability>>).

The accession numbers and database should be listed in a formal "Data Availability" section (placed after Materials & Method) that follows the model below (see also < <https://www.embopress.org/page/journal/14693178/authorguide#dataavailability>>). Please note that the Data Availability Section is restricted to new primary data that are part of this study.

Data availability

Additional information on source data and instruction on how to label the files are available

10) Figure legends and data quantification:

- the name of the statistical test used to generate error bars and P values,
- the EXACT p-values,
- the number (n) of independent experiments (please specify technical or biological replicates) underlying each data point,
- the nature of the bars and error bars (s.d., s.e.m.)

- If the data are obtained from n {less than or equal to} 5, show the individual data points in addition to the SD or SEM.

- If the data are obtained from n {less than or equal to} 2, use scatter blots showing the individual data points.

11) Our journal encourages inclusion of *data citations in the reference list* to directly cite datasets that were re-used and obtained from public databases. Data citations in the article text are distinct from normal bibliographical citations and should directly link to the database records from which the data can be accessed. In the main text, data citations are formatted as follows: "Data ref: Smith et al, 2001" or "Data ref: NCBI Sequence Read Archive PRJNA342805, 2017". In the Reference list, data citations must be labeled with "[DATASET]". A data reference must provide the database name, accession number/identifiers and a resolvable link to the landing page from which the data can be accessed at the end of the reference. Further instructions are available at .

12) All Materials and Methods need to be described in the main text using our 'Structured Methods' format. According to this format, the Methods section includes a Reagents and Tools Table (listing key reagents, experimental models, software and relevant equipment and including their sources and relevant identifiers) followed by a Methods and Protocols section describing the methods, ideally using a step-by-step protocol format. The aim is to facilitate adoption of the methodologies across labs. Please download and fill our Reagents and Tools Table template (.docx), which you can find in our author guidelines:

13) As part of the EMBO publication's Transparent Editorial Process, EMBO Reports publishes online a Review Process File to accompany accepted manuscripts. This File will be published in conjunction with your paper and will include the referee reports, your point-by-point response and all pertinent correspondence relating to the manuscript.

Kind regards,

Martina

REBUTTAL TO REVIEWER COMMENTS

Reviewer #1

In this manuscript, Mitchell et al examine the impact of CHD3 KO (or het) on iPSC differentiation to CNCCs to model how pathogenic CHD3 mutations promote a neurodevelopmental disorder. The authors perform genomic characterization of the KO and het mutants during this differentiation model, and identify loss of CHD3 results in BMP target gene repression and increased mesoderm fate. Finally, the authors attempt to rescue the observed defects by titrating Wnt levels and observe partial rescue. I think the paper is easy to follow, contains interesting data, and establishes a novel role for CHD3 in CNCC differentiation, which may have implications in the disorder highlighted. I have the following suggestions:

We are thankful to the Reviewer for the positive comments on our manuscript. We were able to address ALL the suggestions, which certainly improved the manuscript.

1. Figure 1 presents nice confirmation of the CHD3 KO cell lines being used. However, given that these cell lines were previously published, I suggest moving these data to the supplement.

As recommended, we have moved these data to the supplements (Appendix Figure S1).

2. In the results section for Figure 1, the authors discuss the CHD3 heterozygotes, but I only see the KO cell line data presented. It would be especially nice to see the protein levels of Chd3 in the het.

As recommended, we have performed the Western Blot of the heterozygous line, which can be found in Appendix Figure S1. The blot shows that the heterozygous line has roughly half of the CHD3 protein compared to the wild-type control.

3. The authors discuss which genes are up and downregulated in the Chd3 KO D18 RNAseq, and show a clear heatmap in Figure 2A for WT cells. The same heatmap for candidate genes discussed in the results would be appreciated for Chd3 KO. Only a subset of markers are shown in Fig 2C.

As recommended, we have added the CHD3-KO expression data to heatmap in former Fig. 2A, which is now Fig. 1C.

4. In general 2-3 replicates are presented. While the authors are showing heatmaps for selected locations for individual clones, which is appreciated (ex: Figure 4B and Fig 6), the QC for data quality is missing. For example, show spearman correlation across the genome for datasets as a supplement.

As recommended, we have performed spearman correlations for all the replicates of all the genomic experiments. They can be found in Appendix Figure S3,6,7,8,10. Importantly, all the correlations are statistically significant and with very high rho values, generally above 0.9, supporting high quality of replicated experiments.

5. In the section discussing the results presented in Figure 4, the authors discuss the ATAC-seq peak number changes and overlap with gene expression changes. However, the overlap with gene expression changes is not shown. Making a simple venn diagram would help readers.

As recommended, we have added the Venn Diagrams to former Fig. 4 (now Fig. 3).

a. In addition, showing a heatmap for unchanged ATACseq peaks can help to demonstrate the increase/decrease.

As recommended, we have added the heatmap for unchanged ATAC regions. They can be found in Appendix Figure S6.

6. In Figure 6, the authors present ChIPseq data for CHD3 in D14 and D18 samples, focusing on locations losing or gaining accessibility. What is enrichment at unchanged sites? Is CHD3 specifically enriched at changed locations? Then what about over genes with altered gene expression vs not changed? Is CHD3 only bound to distal elements? Performing an analysis of the peak distribution, perhaps with ChromHMM or other methods to look at promoter vs enhancer vs other locations. These types of analyses could really enrich the interpretation of direct CHD3 function.

As recommended, we have added the heatmaps of CHD3 ChIP-seq at unchanged regions (i.e. regions where the chromatin accessibility is comparable between CHD3-WT and CHD3-KO samples), which revealed little to no binding at CHD3 at these regions. This further supports that the regions that have accessibility alterations upon CHD3 loss are normally bound and regulated by CHD3, while those with no accessibility changes are not regulated by CHD3.

Moreover, as suggested by the Reviewer, we have added a breakdown of the peak distribution, also interpolating them with H3K27ac data, which confirm that most of the regions are distal and likely enhancers.

7. Given the changes in the CHD3 KO accessibility are mostly gene distal, are there existing Hi-C/microC/promoter CaptureC or other that can be used to ask if these are interacting with the predicted genes?

Unfortunately to our knowledge, these types of essays have not been performed genome-wide in CNCCs or in specifying CNCCs.

8. Are the results observed NuRD-based or CHD3 NuRD independent functions? Looking at other NuRD subunit binding or effects in differentiation would help to dig into this a bit more. I realize this is a bit of a big ask, so I am not asking for everything. Are there existing binding data in CNCCs for a NuRD subunit that could be examined for overlap in where these changes occur, for example? I want to be clear I am not asking the authors to do all the experiments for an alternative NuRD subunit.

We thank the Reviewer for this great suggestion. We have performed ChIP-seq of Mbd3 (core NuRD subunit), which revealed very limited overlap between NuRD and CHD3 (~10% of the Mbd3 peaks overlap a CHD3 peak and vice-versa). This may suggest that most of the NuRD complex active at this developmental stage might contain CHD4 and not CHD3, and that CHD3 might work in a NuRD-independent fashion. However, we think that more biochemical work would be needed to support that CHD3 works independently of NuRD (albeit this has been demonstrated for CHD4), so we have stressed this limitation in the manuscript (both results and discussion section).

9. The authors observe defects in CNCCs through genomic experiments. It would be really nice to perform simple wound healing/scratch assays and/or transwell assays to test if the CNCC migration phenotype is reduced in the CHD3 KO as well which would support the transcriptomic data.

As recommended, we have performed a scratch assay, which showed very moderate differences between CHD3-KO and CHD3-WT cells. There could be many potential explanations for this, as some populations of mesodermal cells are also migratory. The scratch assay DATA can be found in Appendix Fig. S5.

10. Related to the above, I am not sure if there is a phenotypic test for enhanced mesoderm. I suspect only IF/expression and morphology are possible, which the authors did. However, sorting the cells (with some defined markers) to ask how many are mesoderm-like vs CNCC in WT vs CHD3 KO would give some information outside of the bulk expression data.

The immunofluorescence data presented in the original version of the manuscript clearly show that all the cells are strongly brachyury positive, suggesting that the mesodermal fate is pervasive and that nearly all the cells follow that trajectory. See updated Fig. 2E.

11. I did not see a reviewer token for the GEO data, so I could not check the deposited datasets.

Sorry for the inconvenient. However, the data are already accessible, so reviewer token is not necessary.

Minor points

12. 1A seems to fit better with Figure 2. Done, thanks for the suggestion.

13. The authors say that the KO cell lines are not defective in pluripotency, but Figures 1G suggests a slight decrease in SSEA-1. Is this reproducibly observed? It is not reproducibly observed, nor it is statistically significant.

14. Would be nice to show number of up and downregulated genes in volcano plots for fast viewing of readers (ex: Fig 2B). Done, thanks for the suggestion.

15. Is it fair to use violin plots when data points are only 2-3 replicates (as in Figures 2C, 3D) Since we also show the actual data point, we personally see no issue in using the violin-plot format. Nonetheless, we have remove them and just left the dot plots.

16. The labels in Fig 4A and 5E are very hard to read. Apologies for this. We have updated the figures improving readability.

17. For browser tracks, the authors show very zoomed in examples (Fig 4C, and especially Fig 6C). showing a bit more of the area around these peaks would give readers a more clear appreciation of the data. We are now showing a broader area and also added RNA-seq data next to it.

18. Related to browser tracks, including more information just as including the gene expression changes (such as in Fig 6C) to enhance the interpretation of the impact of Chd3 binding, accessibility change and then, I presume, reduced Sox9 expression. Similar suggestion for Figure 4C, where I anticipate coordinate transcription changes of the associated genes. See point 17.

19. Do the authors observe any clone variability between the two CHD3 KO clones? There is variability I see in some of the heatmaps, but don't know if that it is because of clones or technical variation. No, we do not observe any significant variability between clones.

Referees cross-commenting

I think that the other reviewer and I are in line with each other in terms of our reviews and thoughts on the manuscript, so I do not have anything to add.

Reviewer #1 (Significance (Required)):

The paper presented by Mitchell et al represents a new role for CHD3 in regulating CNCC differentiation and perhaps explains why CHD3 mutations exist in neurodevelopmental disorders such as Snijders Blok-Campeau Syndrome. Limitations are the reliance on genomic datasets and modeled differentiation, although this permits for more mechanistic studies.

I believe the fields of neural development, stem cell, chromatin biology, and others will be interested in this manuscript.

Thanks so much for your kind words of appreciation of our manuscript.

Reviewer #2

In this manuscript, Mitchell et al. study the function of CHD3 - a member of the NuRD chromatin remodeling complex - during human cranial neural crest cells (CNCC)

differentiation in vitro. To this end they use iPSC CHD3-KO lines. They first observed this deletion has no impact on pluripotency levels of mutant iPSC neither on their capacity to differentiate into the three germ layers derivatives. Yet, they showed these mutant cells are unable to form CNCC as they fail to induce EMT genes and undergo CNCC differentiation. Using ATACseq, the authors then showed CHD3 KO present a reorganization of the chromatin accessible landscape, biasing these cells from a CNCC to a mesoderm fate. They further determine that upon differentiation of CHD3 KO cells, BMP responsive regulatory elements are aberrantly closed, making the cells insensitive to the signaling, explaining how they then fail to generate CNCC. Using ChIP-seq, they confirmed a direct action of CHD3 in making these elements accessible as it normally binds to these chromatin regions to allow proper differentiation. In addition, they demonstrate these BMP responsive genes are bound by DLX5, a transcription factor essential for neural crest development. Finally, the authors showed that during CNCC differentiation, CHD3 KO cells experience an imbalance between BMP and WNT signaling, leading to these cells adopting a mesoderm instead of a CNCC identity. They finally, showed this can be partially rescued by reducing the amount of Wnt signaling - that decreases the mesoderm gene expression - however, it not sufficient to induce a neural crest identity.

We are thankful to the Reviewer for the positive comments on our manuscript. We were able to address ALL the suggestions, which certainly improved the manuscript.

Major comments

1. What is the expression level of CHD3 in the heterozygote line? Does the remaining allele compensate for the loss which will explain the absence of phenotype?

As recommended also by the other Reviewer, we have performed the Western Blot of the heterozygous line, which can be found in Appendix Figure S1. The blot shows that the heterozygous line has roughly half of the CHD3 protein compared to the wild-type control. Hence, we believe that the expression from the wild-type allele is sufficient to explain the absence of phenotype.

2. Since CHD3 shows a progressive upregulation in expression during CNCC differentiation (Fig. 2E), one hypothesis can be that it is not necessary involved in the activation of the CNCC programs but instead it is involved in maintaining these programs active - by keeping regulatory elements accessible. Thus, authors should check expression of CNCC markers, and EMT genes at the same time point than Fig. 2E in both WT and KO cells.

As suggested, we have performed time course RT-qPCR for CNCC markers and EMT markers. The results are in Appendix Figure S4. These data confirm that CNCC markers get activated to a significantly lower extent than the wild-type, suggesting that it is not a matter of maintaining the program active, but to induce them. On the other hand, the EMT markers suggest upregulation of epithelial marker EPCAM and downregulation of VIM only at later timepoints (i.e. after CHD3 is normally upregulated).

3. The authors should use the term "regulatory elements" instead of "enhancers" as they can act either as activator or repressors.

We thank the Reviewer for this suggestion, we have re-worded accordingly.

4. On the same line, while the authors indicate "Motif analysis of the enhancers aberrantly active in CHD3-KO cells ", they haven't shown these are active. They should say they perform the analysis on regulatory elements aberrantly accessible in CHD3 KO.

See point 3, we have re-worded accordingly.

5. The rationale that led the authors to focus on genes typically expressed in the primitive streak and in the early pre-migratory mesoderm, and BMP responsive transcription factors could be better explained. Are they part of the most deregulated genes in the RNAseq analysis?

Thanks for this comment. Not only mesodermal genes are among the most upregulated genes in the RNA-seq, but the motifs for the transcription factors encoded by these genes (e.g. TBR2, Brachyury, GATA, TBX3, TBX6) are among the most frequently represented in the aberrantly accessible cis-regulatory elements. The same applies to BMP responsive factor, such as DLX and MSX paralogs, but the other way around (they are downregulated and enriched in the aberrantly closed ATAC-seq regions).

6. In the absence of CHD3, BMP response is not effective. While the authors nicely showed this is linked with changes in chromatin accessibility, it is necessary to check the expression levels of BMP receptors in CHD3 KO cells.

Thanks for this suggestion. We have checked the expression of these genes, and they were not differentially expressed. This is consistent with the downstream response being affected rather than ligand binding to the receptors.

7. Aberrant early mesoderm signature of the CHD3-KO cells needs to be better shown. It is not obvious from the GO analysis in Fig. 2 and the authors then showed expression of some markers but it is unclear how they picked them up.

See point 5: not only mesodermal genes are among the most upregulated genes in the RNA-seq, but the motifs for the transcription factors encoded by these genes (e.g. TBR2, Brachyury, GATA, TBX3, TBX6) are among the most frequently represented in the aberrantly accessible cis-regulatory elements. See for example expression levels of typical mesodermal genes below:

EOMES – upregulated log₂FC: 5.5

TBXT – upregulated log₂FC: 4.6

MESP1 – upregulated log₂FC: 4.7

MIXL1 – upregulated log₂FC: 5.4

TBX6 – upregulated log₂FC: 3.2

MSGN1 – upregulated log₂FC: 4.6

HAND1 – upregulated log₂FC: 5.5

8. It has been shown that CNCC regulatory elements controlling differentiation genes are primed/accessible prior migration (PMID: 31792380; PMID: 33542111). Since the authors claim "CHD3 may have the role of priming the developing CNCCs to respond to BMP by opening the chromatin at the BMP responsive enhancers", it will be good to perform ATACseq at several time points during the differentiation process to assess the dynamic of chromatin reorganization to see when the switch to mesoderm fate occurs and how accessibility of BMP responsive element changes in WT and KO cells during CNCC differentiation to be able to demonstrate the KO fail to make BMP responsive element accessible or whether it is a defect in the maintenance of this accessibility.

As recommended by the Reviewer we have performed time-course ATAC-seq. Interestingly, this experiment (which can be found in Appendix Figure S9) shows that these regions are accessible from the beginning (day-0) and remain accessible all the way to day-18.

However, in CHD3-KO cells the accessibility drops exactly after day-10, which is the day in which CHD3 is upregulated in wild-type conditions. This suggests that, at least at the chromatin accessibility level, CHD3's role is to maintain these regions accessible. We have clarified this in the manuscript, thanks for the insightful suggestion!

9. The authors claim CHD3 directly binds at BMP responsive enhancers, but in the figure, they show the data for all the region gaining or losing activity. It will be nice to add the information for the BMP responsive elements only.

As recommended, we have added an heatmap for BMP responsive regions only, clearly showing that CHD3 binds them (Supplementary Figure S7).

10. Motifs enrichment analysis of regions gaining accessibility in CHD3 KO do not seem to

be labeled as Wnt responsive elements. The authors need to support better that CHD3 KO express more Wnt signaling/activity.

We have checked expression of many genes that are typically Wnt responsive during mesoderm specification (see also point 7). These include:

EOMES – upregulated log₂FC: 5.5

TBXT – upregulated log₂FC: 4.6

MESP1 – upregulated log₂FC: 4.7

MIXL1 – upregulated log₂FC: 5.4

TBX6 – upregulated log₂FC: 3.2

MSGN1 – upregulated log₂FC: 4.6

HAND1 – upregulated log₂FC: 5.5

These data clearly support that the Wnt-mediated mesodermal program is markedly upregulated.

11. OPTIONAL: Does increasing BMP concentration early during CHD3 KO differentiation has a better effect at rescuing CNCC differentiation?

We were not able to test this (as it was labelled by the reviewer as “optional”).

12. OPTIONAL: While this is not necessary for the current study, it is very intriguing that other CHD family member do not compensate. How this tissue or DNA sequence activity is achieved could be discussed. What are CHD4 or CHD5 expressed during CNCC differentiation? Could they be used to rescue the CHD3 KO phenotype? While this may be difficult to test, it could perhaps be discussed.

As recommended, we have discussed this topic in the discussion, thanks for the suggestion!

Minor comments

13. In the discussion, the authors could indicate whether CHD3 mutants somehow phenocopies some of the craniofacial defects observed in DLX5 mutant patients. **Done.**

14. It is not indicated were to find the data regarding expression epithelial and mesenchymal genes in the CHD3-KO cells. **They are in the heatmap in Fig. 1C.**

15. Authors could add in the discussion what is known about how CHD3 function changes from opening or closing chromatin is very intriguing a could be discussed. **To our knowledge, nothing is known on this. CHD3 is significantly understudied. However, we have mentioned this in the discussion, as suggested.**

Reviewer #2 (Significance (Required)):

Significance

General assessment:

The link between chromatin remodelers and craniofacial defects has been shown in several studies in the past, but it still remains unclear how mutation of a given factor leads to such tissue specific defects. This manuscript represents an interesting and detailed mechanistic study on the role of chromatin remodeler in cell fate decision, showing that reorganization of chromatin accessibility is essential to proper response to signaling pathway and cell differentiation.

Advance:

The authors manage to link how mutant-induced changes in chromatin accessibility biased the cells towards a mesoderm fate as they directly impact the capacity of the cells to respond to BMP signaling - these regions closing upon CHD3 loss. However, the question remains to know whether CHD3 acts as an initiating factor or instead is involved in maintaining these programs active. As noted by the authors, a clinical link (with patient-derived iPCS) would be of great interest but as it stands the story already provide a good mechanistic understanding on how CHD3 control CNCC differentiation.

Audience:

This manuscript will be of great interest for specialized audience, yet a broader public may find it interesting too.

Thanks so much for your kind words of appreciation of our manuscript.

Reviewer field of expertise:

Neural crest and craniofacial development, epigenetics, transcriptomics

Dear Marco,

Thank you for the submission of your revised manuscript to EMBO reports. I had already sent you the referee reports via e-mail and I copy them again below my signature.

Both referees support publication after some minor modifications to clarify text and figures.

From the editorial side, there are also a few things that we need before we can proceed with the official acceptance of your study:

- Summary should be Abstract

- The manuscript sections should be in the following order: Title page - Abstract & Keywords - Introduction - Results - Discussion - Methods - Data Availability - Acknowledgments - Disclosure Statement & Competing Interests - References - Figure Legends - (Main Tables with legends if applicable) - Expanded View Figure Legends.

- Please reduce the number of keywords to 5.

- Please cite the preprint from den Hoed J et al like this:

In-text citation: (preprint: den Hoed et al, 2024)

Reference list: den Hoed J, Wong MMK, Claassen WJJ, de Hoyos L, Lütje L, Heide M, Huttner WB & Fisher SE (2024) The chromatin remodeler CHD3 is highly expressed in mature neurons and regulates genes involved in synaptic development and function. bioRxiv: 2024.04.29.591720 [PREPRINT]

- Regarding the Author Contributions, we now use CRediT to specify the contributions of each author in the journal submission system. Therefore, please remove the Author Contributions from the manuscript file and make sure that the author contributions in our online manuscript tracking system are correct and up-to-date. The information you specified in the system will be automatically retrieved and typeset into the article. You can enter additional information in the free text box provided, if you wish.

- Please add callouts for Figure 3F, Appendix Fig. S3, Appendix Fig. S6, Appendix Fig. S7, Appendix Fig. S8 in the text, wherever appropriate.

- You indicate in the Author Checklist that you authenticated the cell lines, or tested for mycoplasma contamination (C54), but I could not find the relevant statement in the Methods section.

- In the figure legends: please specify whether the replicates are biological, independent replicates or technical replicates for all mentions of "n = #" (also in the Appendix).

- Please indicate the statistical test used for data analysis in the legend of Figure 4B.

- The scale bars are sometimes not well visible, e.g., in Appendix Fig. S1D, G.

- Appendix Fig. S7, S8, S9G: please specify the statistical test used.

- As a standard procedure we edit title and abstract to make it more accessible to our general readership. I have introduced only a minor modification in the abstract and suggested a shorter title. Please review my proposals.

- Doing a spot check on the source data it seems that the images supplied for Figure 1E are a single channel (DAPI) only? I opened them in Fiji, which could also detect one channel only.

The same applies to Figure 2D. Could you please check these and related Source Data?

- We need the Source Data files as one individual ZIP folder per figure, please.

- Finally, EMBO Reports papers are accompanied online by

A) a short (1-2 sentences) summary of the findings and their significance,

B) 2-3 bullet points highlighting key results and

C) a schematic summary figure that provides a sketch of the major findings (not a data image).

Please provide the summary figure as a separate file in PNG or JPG format at a size of 550x300-600 pixels (width x height). Please note that the size is rather small and that text needs to be readable at the final size. Please send us this information along with the revised manuscript.

With kind regards,

Martina

=====

Referee #1:

The authors have addressed all my concerns.

Referee #2:

Altogether, this is a convincing a nicely organized study. While the authors did address all my comments, there are still some minor points that I'd like them to correct in the text to clarify some aspects as indicated below:

1. The abstract is a bit confusing as currently written since the authors start by saying "To unveil the role of CHD3 in craniofacial development, we deleted CHD3 from human iPSC" and then continue by writing "We find that CHD3 is upregulated in early stages of CNCC specification ", which obviously was not done with the mutant cell line.
2. The authors should precise whether in iPSC stage, the 25 differentially expressed genes between CHD3-WT and CHD3-HET-KO and the 62 genes between CHD3-WT and CHD3-KO are the same or not.
3. In Fig.S1C authors observed a complete loss of CHD3 at the protein level in the CHD3-KO iPSCs but then in Fig. 1E, the IF shows some expression of CHD3. Is this because of its upregulation during CNCC differentiation protocol?
4. Authors should flip Fig1E et 1F as the rest of the figures reads left to right
5. Fig. 3 shows "loss of CHD3 had a significant impact on chromatin accessibility at distal cis-regulatory elements". However, the authors need to indicate whether this is because overall the chromatin in CHD3KO is less accessible or if it is an aberrant change in the accessible loci?
6. Authors should refer to surface ectoderm and not non-neural ectoderm as it is not correct to define a structure by something it is not

=====

Suggested Title and Abstract:

The NuRD component CHD3 promotes BMP signalling during cranial neural crest cell specification

Pathogenic genetic variants in the NuRD component CHD3 cause Snijders Blok-Campeau Syndrome, a neurodevelopmental disorder manifesting with intellectual disability, and craniofacial anomalies. To investigate the role of CHD3 in craniofacial development, we differentiated control and *CHD3*-depleted human induced pluripotent stem cells into cranial neural crest cells (CNCCs). We find that CHD3 is upregulated in early stages of CNCC specification in control cells, where it enhances the BMP signalling response by opening the chromatin at BMP-responsive cis-regulatory elements and by increasing the expression of BMP responsive transcription factors, including DLX paralogs. CHD3 loss leads to repression of BMP target genes and loss of chromatin accessibility at cis-regulatory elements usually bound by BMP-responsive factors, causing an imbalance between BMP and Wnt signalling. Consequently, CNCC specification fails, replaced by aberrant early-mesoderm identity, which can be partially rescued by titrating Wnt levels. Our findings highlight a novel role for CHD3 as a pivotal regulator of BMP signalling,

essential for proper neural crest specification and craniofacial development. Moreover, these results suggest a molecular mechanism for the craniofacial anomalies of Snijders Blok-Campeau Syndrome.

Referee #1:

The authors have addressed all my concerns.

Referee #2:

Altogether, this is a convincing and nicely organized study. While the authors did address all my comments, there are still some minor points that I'd like them to correct in the text to clarify some aspects as indicated below:

1. The abstract is a bit confusing as currently written since the authors start by saying "To unveil the role of CHD3 in craniofacial development, we deleted CHD3 from human iPSC" and then continue by writing "We find that CHD3 is upregulated in early stages of CNCC specification ", which obviously was not done with the mutant cell line.
2. The authors should precise whether in iPSC stage, the 25 differentially expressed genes between CHD3-WT and CHD3-HET-KO and the 62 genes between CHD3-WT and CHD3-KO are the same or not.
3. In Fig.S1C authors observed a complete loss of CHD3 at the protein level in the CHD3-KO iPSCs but then in Fig. 1E, the IF shows some expression of CHD3. Is this because of its upregulation during CNCC differentiation protocol?
4. Authors should flip Fig1E et 1F as the rest of the figures reads left to right
5. Fig. 3 shows "loss of CHD3 had a significant impact on chromatin accessibility at distal cis-regulatory elements". However, the authors need to indicate whether this is because overall the chromatin in CHD3KO is less accessible or if it is an aberrant change in the accessible loci?
6. Authors should refer to surface ectoderm and not non-neural ectoderm as it is not correct to define a structure by something it is not

Rev_Com_number: N/a

New_manu_number: EMBOR-2025-61878V2

Corr_author: TRIZZINO

Title: The NuRD component CHD3 regulates BMP signalling response in cranial neural crest cell specification

All editorial and formatting issues were resolved by the authors.

MARCO TRIZZINO
Imperial College London
Department of Life Sciences
Imperial College Road
London SW72AZ
United Kingdom

Dear Marco,

I am very pleased to accept your manuscript for publication in the next available issue of EMBO reports. Thank you for your contribution to our journal.

Kind regards,

Martina
